# Reinforced Genetic Algorithm for Structure-based Drug Design

**Tianfan Fu**[1*], **Wenhao Gao**[2*], **Connor W. Coley**[2,3], **Jimeng Sun**[4,5],
[1]Department of Computational Science and Engineering, Georgia Institute of Technology,
[2]Department of Chemical Engineering, Massachusetts Institute of Technology,
[3]Department of Electrical Engineering and Computer Science, Massachusetts Institute of Technology,
[4] Department of Computer Science, University of Illinois at Urbana-Champaign,
[5] Carle Illinois College of Medicine, University of Illinois at Urbana-Champaign,
[*]Equal Contributions

tfu42@gatech.edu, {whgao,ccoley}@mit.edu, jimeng@illinois.edu

## Abstract

Structure-based drug design (SBDD) aims to discover drug candidates by finding molecules (ligands) that bind tightly to a disease-related protein (targets), which is the primary approach to computer-aided drug discovery. Recently, applying deep generative models for three-dimensional (3D) molecular design conditioned on protein pockets to solve SBDD has attracted much attention, but their formulation as probabilistic modeling often leads to unsatisfactory optimization performance. On the other hand, traditional combinatorial optimization methods such as genetic algorithms (GA) have demonstrated state-of-the-art performance in various molecular optimization tasks. However, they do not utilize protein target structure to inform design steps but rely on a random-walk-like exploration, which leads to unstable performance and no knowledge transfer between different tasks despite the similar binding physics. To achieve a more stable and efficient SBDD, we propose Reinforced Genetic Algorithm (`RGA`) that uses neural models to prioritize the profitable design steps and suppress random-walk behavior. The neural models take the 3D structure of the targets and ligands as inputs and are pre-trained using native complex structures to utilize the knowledge of the shared binding physics from different targets and then fine-tuned during optimization. We conduct thorough empirical studies on optimizing binding affinity to various disease targets and show that `RGA` outperforms the baselines in terms of docking scores and is more robust to random initializations. The ablation study also indicates that the training on different targets helps improve the performance by leveraging the shared underlying physics of the binding processes. The code is available at `https://github.com/futianfan/reinforced-genetic-algorithm`.

## 1 Introduction

Rapid drug discovery that requires less time and cost is of significant interest in pharmaceutical science. Structure-based drug design (SBDD) [1] that leverages the three-dimensional (3D) structures of the disease-related proteins to design drug candidates is one primary approach to accelerate the drug discovery processes with physical simulation and data-driven modeling. According to the lock and key model [2], the molecules that bind tighter to a disease target are more likely to expose bioactivity against the disease, which has been verified experimentally [3]. As AlphaFold2 has provided accurate predictions to most human proteins [4, 5], SBDD has a tremendous opportunity to discover new drugs for new targets that we cannot model before [6].

36th Conference on Neural Information Processing Systems (NeurIPS 2022).

SBDD could be formulated as an optimization problem where the objective function is the binding affinity estimated by simulations such as docking [2]. The most widely used design method is virtual screening, which exhaustively investigates every molecule in a library and ranks them. Lyu et al. successfully discovered new chemotypes for AmpC $\beta$-lactamase and the $D_4$ dopamine receptor by studying hundreds of millions of molecules with docking simulation [7]. However, the number of the drug-like molecules is large as estimated to be $10^{60}$ [1], and it is computationally prohibitive to screen all of the possible molecules. Though machine learning approaches have been developed to accelerate screening [8, 9], it is still challenging to screen large enough chemical space within the foreseeable future.

Instead of naively screening a library, designing drug candidates with generative models has been highlighted as a promising strategy, exemplified by [10, 11]. This class of methods models the problem as the generation of ligands conditioned on the protein pockets. However, as generative models are trained to learn the distribution of known active compounds, they tend to produce molecules similar to training data [12], which discourages finding novel molecules and leads to unsatisfactory optimization performance.

A more straightforward solution is a combinatorial optimization algorithm that searches the implicitly defined discrete chemical space. As shown in multiple standard molecule optimization benchmarks [13, 14, 15], combinatorial optimization methods, especially genetic algorithms (GA) [16, 17], often perform better than deep generative models. The key to superior performance is GA's action definition. Specifically, in each generation (iteration), GA maintains a population of possible candidates (a.k.a. parents) and conducts the crossover between two candidates and mutation from a single candidate to generate new offspring. These two types of actions, crossover and mutation, enable global and local traversal over the chemical space, allowing a thorough exploration and superior optimization performance.

However, most GAs select mutation and crossover operations randomly [16], leading to significant variance between independent runs. Especially in SBDD, when the oracle functions are expensive molecular simulations, it is resource-consuming to ensure stability by running multiple times. Further, most current combinatorial methods are designed for general-purpose molecular optimization and simply use a docking simulation as an oracle. It is challenging to leverage the structure of proteins in these methods, and we need to start from scratch whenever we change a protein target, even though the physics of ligand-protein interaction is shared. Ignoring the shared information across tasks leads to unnecessary exploration steps and, thus, demands for many more oracle calls, which require expensive and unnecessary simulations [18].

To overcome these issues in the GA method, we propose Reinforced Genetic Algorithm (`RGA`), which attempts to reformulate an evolutionary process as a Markov decision process and uses neural networks to make informed decisions and suppress the random-walk behavior. Specifically, we utilize an E(3)-equivariant neural network [19] to choose parents and mutation types based on the 3D structure of the ligands and proteins. The networks are pre-trained with various native complex structures to utilize the knowledge of the shared binding physics between different targets and then fine-tuned with a reinforcement learning algorithm during optimizations. We test `RGA`'s performance with various disease-related targets, including the main protease of SARS-CoV-2.

The main contributions of this paper can be summarized as follows:

- We propose an evolutionary Markov decision process (EMDP) that reformulates an evolutionary process as a Markov decision process, where the state is a population of molecules instead of a single molecule (Section 3.2).

- We show the first successful attempt to use a neural model to guide the crossover and mutation operations in a genetic algorithm to suppress random-walk behavior and explore the chemical space intelligently (Section 3.3).

- We present a structure-based de novo drug design algorithm that outperforms baseline methods consistently through thorough empirical studies on optimizing binding affinity by leveraging the underlying binding physics (Section 4).

## 2 Related Works

We will discuss the related works on methods of drug design and discuss the advantage of the proposed method over the existing works.

**General Molecular Design**. Molecular generation methods offer a promising direction for the automated design of molecules with desired pharmaceutical properties such as synthesis accessibility and drug-likeliness. Based on how to generate or search molecules, these approaches can be categorized into two types, (1) deep generative models (DGMs) imitate the molecular data distribution, including variational autoencoder (VAE) [20, 21], generative adversarial network (GAN) [22, 23], normalizing flow model [24, 25], energy based model [26]; and (2) combinatorial optimization methods directly search over the discrete chemical space, including genetic algorithm (GA) [16, 27, 28], reinforcement learning approaches (RL) [29, 30, 31, 32, 33], Bayesian optimization (BO) [34], Markov chain Monte Carlo (MCMC) [35, 36, 37] and gradient ascent [38, 39].

General molecular design algorithms often use general black-box oracle functions, and some are only tested with trivial or self-designed oracles. For example, using penalized octanol-water partition coefficient (LogP) as the oracle function, it grows monotonically with the number of carbons, and thus there exists a trivial policy to optimize LogP. These oracles do not reflect the challenges of real drug discovery, and those algorithms have limited value for pharmaceutical discovery. Recent works are optimizing docking scores to simulate a more realistic discovery scenario [40, 41, 18, 42], same as our work. However, they are still ignoring the information in the given protein structures that could potentially accelerate the design process. However, the extension to leveraging the structural knowledge is nontrivial.

**Structure-based Drug Design**. Structure-based drug design (SBDD) could utilize the structural information to guide the design of molecules, which are potentially more efficient in drug discovery tasks but poses additional challenges of how to leverage the structures. Since early 1990s, various SBDD algorithms have been proposed, mostly based on combinatorial optimization algorithms such as tree search [43, 44, 45] and evolutionary algorithms [46, 47]. Those methods typically optimize the ligands in the pockets according to a physical model characterizing the binding affinity. For example, RASSE [43] used a force-field-like scoring function [48] to evaluate the partial solutions within a tree search. However, obtaining a fast and accurate model to quantify binding free energy itself is still an unsolved challenge.

Recently, generative modeling of 3D molecules conditioned on protein targets is attracting more attention [10, 11]. Similar to DGMs in general molecular design, those methods learn the atom's compositional and spatial distribution of native structure of protein-ligand complexes with neural models and design new ligands by complete the complex structure given targets. Deep generative models are end-to-end and data-driven thus surpass the necessity of understanding the physics of interaction. However, as the training objective is to learn the distribution of known active compounds, the models tend to produce molecules close to the training set [12], which is undesired in terms of patentability and leads to unsatisfactory optimization performance.

## 3 Method

In this paper, we focus on structure-based drug design. The goal is to design drug molecules (a.k.a. ligands) that could bind tightly with the disease-related proteins (a.k.a. targets). Given the 3D structures of the target proteins, including binding site information, docking is a popular computational method for assessing the binding affinity, which can be roughly retrieved as the free energy changes during the binding processes. We present a variant of genetic algorithm that is guided by reinforcement learning and a docking oracle. Next, we will first describe the general evolutionary process used in genetic algorithms (Section 3.1); Then we will present how to model this evolutionary process as a Markov decision process (MDP) where RL framework can be constructed (Section 3.2); After that, we describe the detailed implementation of this MDP framework using multiple policy networks (Section 3.3).

## 3.1 Evolutionary Process

In this section, we introduce the primary setting of the evolutionary processes. With both optimization performance and synthetic accessibility taken into account [49, 14], we follow the action settings in Autogrow 4.0 [17]. It demonstrated superior performance over other GA variants in the empirical validation of structure-based drug design [17], and its mutation actions originated from chemical reactions so that the designed molecules are more likely to be synthesizable. Specifically, an evolutionary processes starts by randomly sampling a *population* of drug candidates from a library. In each *generation* (iteration), it carries out (i) *crossover* between parents selected from the last generation, and (ii) *mutation* on a single child to obtain the offspring pool. An illustration of both crossover and mutation operations is available in Appendix. Note that we only adopted the action settings from Autogrow 4.0, without using other tricks such as elitism.

**Crossover**, also called recombination, combines the structure of two parents to generate new children. Following Autogrow 4.0 [17], we select two parents from the last generation and search for the largest common substructure shared between them. Then we generate two children by randomly switching their decorating moieties, i.e., the side chains attached to the common substructure.

**Mutation** operates on a single parent molecule and modifies its structure slightly. Following Autogrow 4.0 [17], we adopt transformations based on chemical reactions. Unlike naively defined atom-editing actions, mutation steps based on chemical reactions could ensure all modification is reasonable in reality, leading to a larger probability of designing synthesizable molecules. We included two types of chemical reactions: uni-molecular reactions, which only require one reactant, and bi-molecular reactions, which require two reactants. While uni-molecular reactions could be directly applied to the parent, we sample a purchasable compound to react with the parent when conducting a bi-molecular reaction. In both cases, the parent serves as one reactant, and we use the main product as the child molecule. We use the chemical reactions from [17], which was originally from [47, 50].

**Evolution**. At the $t$-th generation (iteration), given a population of molecules denoted as $\mathcal{S}^{(t)}$, we generate an offspring pool denoted as $\mathcal{Q}^{(t)}$ by applying crossover and mutation operations. Then we filter out the ones with undesirable physical and chemical properties (e.g., poor solubility, high toxicity) in the offspring pool and select the most promising $K$ to form the next generation pool ($\mathcal{S}^{(t+1)}$).

## 3.2 Evolutionary Markov Decision Process

Next we propose the evolutionary Markov decision process (EMDP) that formulates an evolutionary process of genetic algorithm as a Markov decision process (MDP). The primary purpose is to utilize reinforcement learning algorithms to train networks to inform the decision steps to replace random selections. Taking a generation as a state, Markov property that requires $P(\mathcal{S}^{(t+1)}|\mathcal{S}^{(1)}, \cdots, \mathcal{S}^{(t)}) = P(\mathcal{S}^{(t+1)}|\mathcal{S}^{(t)})$ is naturally satisfied by the evolutionary process described above, where $\mathcal{S}^{(t)}$ denotes the state at the $t$-th generation, which is the population of ligands. We use $X$ to denote a ligand. We elaborate essential components for Markov decision process as follows, and the EMDP pipeline is illustrated in Figure 1.

**State Space**. We define the population at the $t$-step generation, $\mathcal{S}^{(t)}$, in the evolutionary process as the state at the $t$-step in an EMDP. A state includes population of candidate molecules (i.e., ligand, denoted $X$) and their 3D poses docked to the target, fully observable to the RL agent. At the beginning of the EMDP, we randomly select a population of candidate molecules and use docking simulation to yield their 3D poses as the initial state.

**Action Space**. The actions in an EMDP are to conduct the two evolutionary steps: crossover and mutation, in a population. For each evolutionary step, we need two actions to conduct it. Concretely, crossover ($X_{\text{crossover}}^{\text{parent 1}}, X_{\text{crossover}}^{\text{parent 2}} \xrightarrow{\text{crossover}} X_{\text{crossover}}^{\text{child 1}}, X_{\text{crossover}}^{\text{child 2}}$) can be divided to two steps: (1) select the first candidate ligand $X_{\text{crossover}}^{\text{parent 1}}$ from the current state (population $\mathcal{S}^{(t)}$); (2) conditioned on the first selected candidate $X_{\text{crossover}}^{\text{parent 1}}$, select the second candidate ligand $X_{\text{crossover}}^{\text{parent 2}}$ from the remaining candidate ligand set $\mathcal{S}^{(t)} - \{X_{\text{crossover}}^{\text{parent 1}}\}$ and apply crossover (Section 3.1) to them.

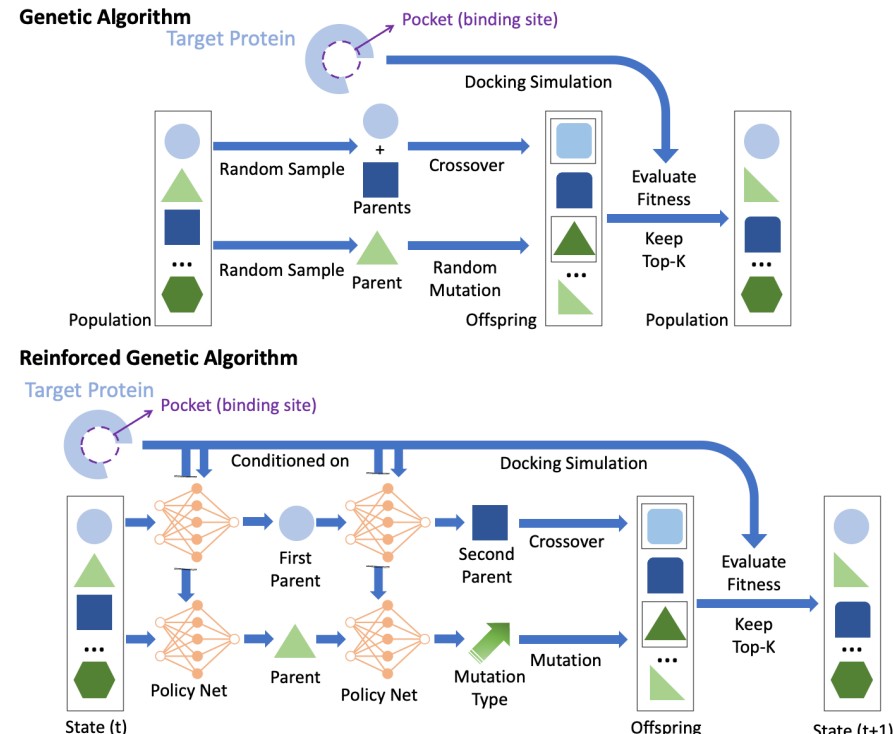

Figure 1: We illustrate one generation (iteration) of GA (top) and RGA pipeline (bottom). Specifically, we train policy networks that take the target and ligand as input to make informed choices on parents and mutation types in RGA.

Mutation ($X_{\text{mutation}}^{\text{parent}} \xrightarrow{\text{mutated by } \xi} X_{\text{mutation}}^{\text{child}}$) can be divided to two steps: (1) select the candidate ligand $X_{\text{mutated}}^{\text{parent}}$ to be mutated from the current state (population $\mathcal{S}^{(t)}$); (2) conditioned on the selected candidate ligand $X_{\text{mutated}}^{\text{parent}}$, select the reaction $\xi$ from the reaction set $\mathcal{R}$ and apply it to $X_{\text{mutated}}^{\text{parent}}$.

As applying the crossover and mutation steps are deterministic, the actions in an EMDP focus on selecting parents and mutation types. Upon finish the action, we could obtain offspring pool, $\mathcal{Q}^{(t)}$.

**State Transition Dynamics**. The state transition in an EMDP is identical to the evolution in an evolutionary process. Once we finish the actions and obtain the offspring pool, $\mathcal{Q}^{(t)} = \{X^{\text{child } 1}, X^{\text{child } 2}, \cdots\}$, we apply molecular quality filters to filter out the ones unlikely to be drug and then select the most promising $K$ to form the parent set for the next generation ($\mathcal{S}^{(t+1)}$).

**Reward**. We define the reward as the binding affinity change (docking score). The actions leading to stronger binding score would be prioritized. As there is no "episode" concept in an EMDP, we treat every step equally.

### 3.3 Target-Ligand Policy Network

To utilize molecular structures' translational and rotational invariance, we adopt equivariance neural networks (ENNs) [19] as the target-ligand policy neural networks to select the actions in both mutation and crossover steps. Each ligand has a 3D pose that binds to the target protein, and the complex serves as the input of ENN.

Specifically, we want to model a 3D graph $\mathcal{Y}$, which can be ligand, target, or target-ligand complex. The input feature can be described as $\mathcal{Y} = (\mathcal{A}, \mathcal{Z})$, where $\mathcal{A}$ represents atoms' categories (the vocabulary set $\mathcal{V} = \{H, C, O, N, \cdots\}$) and $\mathcal{Z}$ represents 3D coordinates of the atoms. Suppose $\mathbf{D} \in \mathbb{R}^{|\mathcal{V}| \times d}$ is the embedding matrix of all the categories of atoms in a vocabulary set $\mathcal{V}$, is randomly initialized and learnable, $d$ is the hidden dimension in ENN. Each kind of atom corresponds to a row in $\mathbf{D}$. We suppose there are $N$ atoms, and each atom corresponds to a node in the 3D graph. Node embeddings at the $l$-th layer are denoted as $\mathbf{H}^{(l)} = \{\mathbf{h}_i^{(l)}\}_{i=1}^N$, where $l = 0, 1, \cdots, L$, $L$ is number of layers in ENN. The initial node embedding $\mathbf{h}_i^{(0)} = \mathbf{D}^\top \mathbf{a}_i \in \mathbb{R}^d$ embeds the $i$-th node, where

$\mathbf{a}_i$ is one-hot vector that encode the category of the $i$-th atom. Coordinate embeddings at the $l$-th layer are denoted $\mathbf{Z}^{(l)} = \{\mathbf{z}_i^{(l)}\}_{i=1}^N$. The initial coordinate embeddings $\mathbf{Z}^{(0)} = \{\mathbf{z}_i\}_{i=1}^N$ are the real 3D coordinates of all the nodes. The following equation defines the feedforward rules of ENN, for $i, j = 1, \cdots, N$, $i \neq j$, $l = 0, 1, \cdots, L-1$, we have

$$\mathbf{w}_{ij}^{(l+1)} = \mathrm{MLP}_e\left(\mathbf{h}_i^{(l)} \oplus \mathbf{h}_j^{(l)} \oplus \|\mathbf{z}_i^{(l)} - \mathbf{z}_j^{(l)}\|_2^2\right) \in \mathbb{R}^d, \qquad \mathbf{v}_i^{(l+1)} = \sum_{j=1,j\neq i}^N \mathbf{w}_{ij}^{(l+1)} \in \mathbb{R}^d,$$

$$\mathbf{z}_i^{(l+1)} = \mathbf{z}_i^{(l)} + \sum_{j=1,j\neq i}^N \left(\mathbf{z}_i^{(l)} - \mathbf{z}_j^{(l)}\right)\mathrm{MLP}_x\left(\mathbf{w}_{ij}^{(l)}\right) \in \mathbb{R}^3, \quad \mathbf{h}_i^{(l+1)} = \mathrm{MLP}_h\left(\mathbf{h}_i^{(l)} \oplus \mathbf{v}_i^{(l+1)}\right) \in \mathbb{R}^d,$$

$$\mathbf{h}_{\mathcal{Y}} = \sum_{i=1}^N \mathbf{h}_i^{(L)} \in \mathbb{R}^d \quad \Longrightarrow \quad \underline{\mathbf{h}_{\mathcal{Y}} = \mathrm{ENN}(\mathcal{Y})}$$

$$\tag{1}$$

where $\oplus$ denotes the concatenation of vectors; $\mathrm{MLP}_e(\cdot) : \mathbb{R}^{2d+1} \to \mathbb{R}^d; \mathrm{MLP}_x(\cdot) : \mathbb{R}^d \to \mathbb{R}; \mathrm{MLP}_h(\cdot) : \mathbb{R}^{2d} \to \mathbb{R}^d$ are all two-layer multiple layer perceptrons (MLPs) with Swish activation in the hidden layer [51]. At the $l$-th layer, $\mathbf{w}_{ij}^{(l)}$ represents the message vector for the edge from node $i$ to node $j$; $\mathbf{v}_i^{(l)}$ represents the message vector for node $i$, $\mathbf{z}_i^{(l)}$ is the position embedding for node $i$; $\mathbf{h}_i^{(l)}$ is the node embedding for node $i$. $\mathbf{H}^{(L)} = [\mathbf{h}_1^{(L)}, \cdots, \mathbf{h}_N^{(L)}]$ are the node embeddings of the $L$-th (last) layer. We aggregate them using sum function as readout function to obtain a representation of the 3D graph, denoted $\mathbf{h}_{\mathcal{Y}}$. The whole process is written as $\mathbf{h}_{\mathcal{Y}} = \mathrm{ENN}(\mathcal{Y})$.

**Crossover Policy Network**. We design two policy networks for two corresponding actions in a crossover, as mentioned in Section 3.2. (1) the first action in crossover operation is to select the first parent ligand $X_{\mathrm{crossover}}^{\mathrm{parent\ 1}}$ from the population $\mathcal{S}^{(t)}$. Similar to the first action in mutation operation, we obtain a valid probability distribution over all the available ligands based on target-ligand complex as input feature and ENN as the neural network architecture, the selection probability of the ligand $X_{\mathrm{crossover}}^{\mathrm{parent\ 1}} \in \mathcal{S}^{(t)}$ is $p_{\mathrm{crossover}}^{(1)}(X_{\mathrm{crossover}}^{\mathrm{parent\ 1}}|\mathcal{S}^{(t)}) = \frac{\exp\left(\mathrm{MLP}(\mathbf{h}_{\mathcal{T}\&X_{\mathrm{crossover}}^{\mathrm{parent\ 1}}})\right)}{\sum_{X' \in \mathcal{S}^{(t)}} \exp\left(\mathrm{MLP}(\mathbf{h}_{\mathcal{T}\&X'})\right)}$, where $\mathcal{T}$ and $X$ denotes target and ligand (including 3D pose), respectively, $\mathcal{T}\&X$ denotes target-ligand complex. (2) The second action is to select the second parent ligand conditioned on the first parent ligand selected in the first action. Specifically, for ligand in the remaining population set, we concatenate the ENN's embedding of the target, first parent ligand $X_{\mathrm{crossover}}^{\mathrm{parent\ 1}}$ and the second parent ligand $X_{\mathrm{mutation}}^{\mathrm{parent\ 2}}$, and feed it into an MLP to estimate a scalar as an unnormalized probability. The unnormalized probabilities for all the ligands in the remaining population set are normalized via softmax function, i.e., $p_{\mathrm{crossover}}^{(2)}(X_{\mathrm{crossover}}^{\mathrm{parent\ 2}}|X_{\mathrm{crossover}}^{\mathrm{parent\ 1}}, \mathcal{S}^{(t)}) = \mathrm{Softmax}\{\mathrm{MLP}(\mathbf{h}_{\mathcal{T}} \oplus \mathbf{h}_{X_{\mathrm{crossover}}^{\mathrm{parent\ 1}}} \oplus \mathbf{h}_{X_{\mathrm{crossover}}^{\mathrm{parent\ 2}}}), \cdots, \}_{X_{\mathrm{crossover}}^{\mathrm{parent\ 2}} \in \mathcal{S}^{(t)} - \{X_{\mathrm{crossover}}^{\mathrm{parent\ 1}}\}}$. Given two parents ligands, crossover finds the largest substructure that the two parent compounds share and generates a child by combining their decorating moieties. Thus, the generation of child ligands are deterministic, and the probability of the generated ligands $X_{\mathrm{crossover}}^{\mathrm{child}}$ is

$$p_{\mathrm{crossover}}(X_{\mathrm{crossover}}^{\mathrm{child\ 1}}, X_{\mathrm{crossover}}^{\mathrm{child\ 2}}|\mathcal{S}^{(t)}) = p_{\mathrm{crossover}}(X_{\mathrm{crossover}}^{\mathrm{parent\ 1}}, X_{\mathrm{crossover}}^{\mathrm{parent\ 2}}|\mathcal{S}^{(t)})$$
$$= p_{\mathrm{crossover}}^{(1)}(X_{\mathrm{crossover}}^{\mathrm{parent\ 1}}|\mathcal{S}^{(t)}) \cdot p_{\mathrm{crossover}}^{(2)}(X_{\mathrm{crossover}}^{\mathrm{parent\ 2}}|X_{\mathrm{crossover}}^{\mathrm{parent\ 1}}, \mathcal{S}^{(t)}).$$

$$\tag{2}$$

**Mutation Policy Network**. We design two policy networks for two corresponding actions in mutation, as mentioned in Section 3.2. (1) the first action in mutation operation is to select a candidate ligand to be mutated from population $\mathcal{S}^{(t)}$. It models the 3D target-ligand complex to learn if there is improvement space in the current complex. Formally, we obtain a valid probability distribution over all the available ligands based on target-ligand complex as input feature and ENN as neural architecture, the selection probability of the ligand $X_{\mathrm{mutation}}^{\mathrm{parent}} \in \mathcal{S}^{(t)}$ is

$p_{\mathrm{mutation}}^{(1)}(X_{\mathrm{mutation}}^{\mathrm{parent}}|\mathcal{S}^{(t)}) = \frac{\exp\left(\mathrm{MLP}(\mathbf{h}_{\mathcal{T}\&X_{\mathrm{mutation}}^{\mathrm{parent}}})\right)}{\sum_{X' \in \mathcal{S}^{(t)}} \exp\left(\mathrm{MLP}(\mathbf{h}_{\mathcal{T}\&X'})\right)}$, where $\mathcal{T}\&X$ denotes target-ligand complex, $\mathbf{h}_{\mathcal{T}\&X} = \mathrm{ENN}(\mathcal{T}\&X)$ represents the ENN's embedding of target-ligand complex. (2) The second action is to select the SMARTS reaction from the reaction set conditioned on the selected ligand in the first action. Specifically, for each reaction, we generate the new ligand $X_{\mathrm{mutation}}^{\mathrm{child}}$, then

obtain the embedding of target, first ligand $X_{\text{mutation}}^{\text{parent}}$ and the new ligand $X_{\text{mutation}}^{\text{child}}$ through ENN, concatenate these three embeddings and feed it into an MLP to estimate a scalar as unnormalized probability. The unnormalized probabilities for all the reactions are normalized via softmax function, i.e., $p_{\text{mutation}}^{(2)}(\xi|X_{\text{mutation}}^{\text{parent}}, \mathcal{S}^{(t)}) = \text{Softmax}\big\{\text{MLP}(\mathbf{h}_{\mathcal{T}} \oplus \mathbf{h}_{X_{\text{mutation}}^{\text{parent}}} \oplus \mathbf{h}_{X_{\text{mutation}}^{\text{child}}}]), \cdots, \big\}_{\xi \in \mathcal{R}}$, where $X_{\text{mutation}}^{\text{parent}} \xrightarrow{\text{mutated by } \xi} X_{\text{mutation}}^{\text{child}}$, $\mathcal{R}$ is the reaction set. The probability of the generated ligand $X_{\text{mutation}}^{\text{child}}$ is

$$p_{\text{mutation}}(X_{\text{mutation}}^{\text{child}}|\mathcal{S}^{(t)}) = p_{\text{mutation}}^{(1)}(X_{\text{mutation}}^{\text{parent}}|\mathcal{S}^{(t)}) \cdot p_{\text{mutation}}^{(2)}(\xi|X_{\text{mutation}}^{\text{parent}}, \mathcal{S}^{(t)}). \tag{3}$$

**Policy Gradient**. We leverage policy gradient to train the target-ligand policy neural network. Specifically, we consider maximizing the expected reward as objective via REINFORCE [29],

$$\max \ \mathbb{E}_{X \sim p(X|\mathcal{S}^{(t)})}\big[\text{Reward}(X)\big], \tag{4}$$

where $p(X)$ is defined in Equation (2) and (3) for crossover and mutation, respectively. The whole pipeline is illustrated in Figure 1. To provide a warm start and leverage the structural information, we pretrain the ENNs on 3D target-ligand binding affinity prediction task, where the inputs are the target-ligand complexes, and the outputs are their binding affinity.

## 4 Experiment

In this section, we briefly describe the experimental setup and results. The Appendix includes more details, including software configuration, implementation details, dataset description & processing, hyperparameter tuning, ablation study, and additional experimental results. The code is available at `https://github.com/futianfan/reinforced-genetic-algorithm`.

### 4.1 Experimental Setup

**Docking Simulation**. We adopt AutoDock Vina [52] to evaluate the binding affinity. The docking score estimated by AutoDock Vina is called Vina score and roughly characterizes the free energy changes of binding processes in kcal/mol. Thus lower Vina score means a stronger binding affinity between the ligand and target. We picked various disease-related proteins, including G-protein coupling receptors (GPCRs) and kinases from DUD-E [53] and the SARS-CoV-2 main protease [54] as targets. Please see the Appendix for more information.

**Baselines**. The baseline methods cover traditional brute-force search methods (Screening), deep generative models (JTVAE and Gen3D), genetic algorithm (GA+D, graph-GA, Autogrow 4.0), reinforcement learning methods (MolDQN, RationaleRL, REINVENT, GEGL), and MCMC method (MARS). Gen3D and Autogrow 4.0 are structure-based drug design methods, while others are general-purpose molecular design methods. Although methods explicitly utilizing target structures are relatively few, we add general-purpose molecular design methods optimizing the same docking oracle scores as ours, which is a common use case, as baselines [16, 14]. Concretely, **Screening** mimics high throughput screening via sampling from ZINC database randomly; **JTVAE** (Junction Tree Variational Auto-Encoder) [21] uses a Bayesian optimization on the latent space to indirectly optimize molecules; **Gen3D** [10] is an auto-regressive generative model that grows 3D structures atom-wise inside the binding pocket; **GA+D** [27] represents molecule as SELFIES string [55] and uses genetic algorithm enhanced by a discriminator neural network; **Graph-GA** [16] conduct genetic algorithm on molecular graph representation; **Autogrow 4.0** [17] is the state-of-the-art genetic algorithm in structure-based drug design; **MolDQN** (Molecule Deep Q-Network) [31] leverages deep Q-value learning to grow molecules atom-wisely; **RationaleRL** [32] uses rationale (e.g., functional groups or subgraphs) as the building block and a policy gradient method to guide the training of graph neural network-based generator; **REINVENT** [29] represent molecules as SMILES string and uses policy gradient based reinforcement learning methods to guide the training of the RNN generator; **GEGL** (genetic expert-guided learning) [33] uses LSTM guided by reinforcement learning to imitate the GA exploration; **MARS** (Markov Molecule Sampling) [36] leverages Markov chain Monte Carlo sampling (MCMC) with adaptive proposal and annealing scheme to search chemical space. To conduct a fair comparison, we limit the number of oracle calls to 1,000 times for each method. All the baselines can be run with one-line code using the software (`https://github.com/wenhao-gao/mol_opt`) in practical molecular optimization benchmark [15].

Table 1: The summarized performance of different methods. The mean and standard deviation across targets are reported. Arrows (↑, ↓) indicate the direction of better performance. For each metric, the best method is underlined and the top-3 methods are bolded. `RGA`-pretrain and `RGA`-KT are two variants of `RGA` that without pretraining and without training on different target proteins, respectively.

| Method | TOP-100↓ | TOP-10↓ | TOP-1↓ | Nov↑ | Div↑ | QED↑ | SA↓ |
|---|---|---|---|---|---|---|---|
| screening | $-9.351_{\pm0.643}$ | $-10.433_{\pm0.563}$ | $-11.400_{\pm0.630}$ | $0.0_{\pm0.0\%}$ | $0.858_{\pm0.005}$ | $0.678_{\pm0.022}$ | $2.689_{\pm0.077}$ |
| MARS | $-7.758_{\pm0.612}$ | $-8.875_{\pm0.711}$ | $-9.257_{\pm0.791}$ | $100.0_{\pm0.0\%}$ | $\mathbf{0.877_{\pm0.001}}$ | $\mathbf{0.709_{\pm0.008}}$ | $\mathbf{\underline{2.450_{\pm0.034}}}$ |
| MolDQN | $-6.287_{\pm0.396}$ | $-7.043_{\pm0.487}$ | $-7.501_{\pm0.402}$ | $100.0_{\pm0.0\%}$ | $\mathbf{0.877_{\pm0.009}}$ | $0.170_{\pm0.024}$ | $5.833_{\pm0.182}$ |
| GEGL | $-9.064_{\pm0.920}$ | $-9.91_{\pm0.990}$ | $-10.45_{\pm1.040}$ | $100.0_{\pm0.0\%}$ | $0.853_{\pm0.003}$ | $0.643_{\pm0.014}$ | $2.99_{\pm0.054}$ |
| REINVENT | $-10.181_{\pm0.441}$ | $-11.234_{\pm0.632}$ | $-12.010_{\pm0.833}$ | $100.0_{\pm0.0\%}$ | $0.857_{\pm0.011}$ | $0.445_{\pm0.058}$ | $2.596_{\pm0.116}$ |
| RationaleRL | $-9.233_{\pm0.920}$ | $-10.834_{\pm0.856}$ | $-11.642_{\pm1.102}$ | $100.0_{\pm0.0\%}$ | $0.717_{\pm0.025}$ | $0.315_{\pm0.023}$ | $2.919_{\pm0.126}$ |
| JTVAE | $-9.291_{\pm0.702}$ | $-10.242_{\pm0.839}$ | $-10.963_{\pm1.133}$ | $98.0_{\pm0.027\%}$ | $0.867_{\pm0.001}$ | $0.593_{\pm0.035}$ | $3.222_{\pm0.136}$ |
| Gen3D | $-8.686_{\pm0.450}$ | $-9.285_{\pm0.584}$ | $-9.832_{\pm0.324}$ | $100.0_{\pm0.0\%}$ | $\mathbf{0.870_{\pm0.006}}$ | $0.701_{\pm0.016}$ | $3.450_{\pm0.120}$ |
| GA+D | $-7.487_{\pm0.757}$ | $-8.305_{\pm0.803}$ | $-8.760_{\pm0.796}$ | $99.2_{\pm0.011\%}$ | $0.834_{\pm0.035}$ | $0.405_{\pm0.024}$ | $5.024_{\pm0.164}$ |
| Graph-GA | $\mathbf{-10.848_{\pm0.860}}$ | $\mathbf{-11.702_{\pm0.930}}$ | $\mathbf{-12.302_{\pm1.010}}$ | $100.0_{\pm0.0\%}$ | $0.811_{\pm0.037}$ | $0.456_{\pm0.067}$ | $3.503_{\pm0.367}$ |
| Autogrow 4.0 | $\mathbf{-11.371_{\pm0.398}}$ | $\mathbf{-12.213_{\pm0.623}}$ | $\mathbf{-12.474_{\pm0.839}}$ | $100.0_{\pm0.0\%}$ | $0.852_{\pm0.011}$ | $\mathbf{0.748_{\pm0.022}}$ | $\mathbf{2.497_{\pm0.049}}$ |
| RGA (ours) | $\mathbf{\underline{-11.867_{\pm0.170}}}$ | $\mathbf{\underline{-12.564_{\pm0.287}}}$ | $\mathbf{\underline{-12.869_{\pm0.473}}}$ | $100.0_{\pm0.0\%}$ | $0.857_{\pm0.020}$ | $\mathbf{0.742_{\pm0.036}}$ | $\mathbf{2.473_{\pm0.048}}$ |
| RGA - pretrain | $-11.443_{\pm0.219}$ | $-12.424_{\pm0.386}$ | $-12.435_{\pm0.654}$ | $100.0_{\pm0.0\%}$ | $0.854_{\pm0.035}$ | $0.750_{\pm0.034}$ | $2.494_{\pm0.043}$ |
| RGA - KT | $-11.434_{\pm0.169}$ | $-12.437_{\pm0.354}$ | $-12.502_{\pm0.603}$ | $100.0_{\pm0.0\%}$ | $0.853_{\pm0.028}$ | $0.738_{\pm0.034}$ | $2.501_{\pm0.050}$ |

**Dataset**: we randomly select molecules from ZINC [56] database (around 250 thousands drug-like molecules) as 0-th generation of the genetic algorithms (`RGA`, Autogrow 4.0, GA+D). ZINC also serves as the training data for pretraining the model in JTVAE, REINVENT, RationaleRL, etc. We adopt CrossDocked2020 [57] dataset that contains around 22 million ligand-protein complexes as the training data for pretraining the policy neural networks, as mentioned in Section 3.3. More descriptions are available in Appendix.

**Metrics**. The selection of evaluation metrics follows recent works in molecule optimization [21, 27, 32, 36] and structure-based drug design [17, 10, 14]. For each method, we select top-100 molecules with the best docking scores for evaluation and consider the following metrics: **TOP-1/10/100** (average docking score of top-1/10/100 molecules): docking score directly measures the binding affinity between the ligand and target and is the most informative metric in structure-based drug design; **Novelty (Nov)** (% of the generated molecules that are not in training set); **Diversity (Div)** (average pairwise Tanimoto distance between the Morgan fingerprints); We also evaluate some simple pharmaceutical properties, including quantitative drug-likeness (**QED**) and synthetic accessibility (**SA**). QED score indicates drug-likeliness ranging from 0 to 1 (higher the better). SA score ranges from 1 to 10 (lower the better). All the evaluation functions are available at Therapeutics data commons (TDC, `https://tdcommons.ai/fct_overview`) [14, 58].

### 4.2 Results

**Stronger Optimization Performance**. We summarized the main results of the structure-based drug design in Table 1. We evaluate all the methods on all targets and report each metric's mean and standard deviations across all targets. Our result shows `RGA` achieves the best performance in TOP-100/10/1 scores among all methods we compared. Compared to Autogrow 4.0, `RGA`'s better performance in docking score demonstrates that the policy networks contribute positively to the chemical space navigation and eventually help discover more potent binding molecules. On the other hand, including longer-range navigation steps enabled by crossover leads to superior performance than other RL methods (REINVENT, MolDQN, GEGL and RationaleRL) that only focus on local modifications. In addition, we also observed competitive structure quality measured by QED ($> 0.7$) and SA_Score ($< 2.5$) in Autogrow 4.0 and `RGA` without involving them as optimization objectives, thanks to the mutation steps originating from chemical reactions. We visualize two designed ligands with optimal affinity for closer inspection in Figure 2(a) and 2(b), and find both ligands bind tightly with the targets.

**Suppressed Random-Walk Behavior**. Especially in SBDD, when the oracle functions are expensive molecular simulations, robustness to random seeds is essential for improving the worst-case performance of algorithms. One of the major issues in traditional GAs is that they have a significant variance between multiple independent runs as they randomly select parents for crossover and mutation types. To examine this behavior, we run five independent runs for `RGA`, Autogrow 4.0 and graph-GA (three best baselines, all are GA methods) on all targets and plot the standard deviations between runs in Figure 2(c) and 2(d). With policy networks guiding the action steps, we observed that the random-walk behavior in Autogrow 4.0 was suppressed in `RGA`, indicated by the smaller variance.

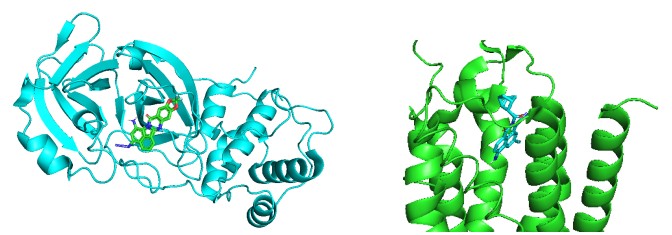

(a) Example of 7l11, -10.8 kcal/mol (b) Example of 3eml, -13.2 kcal/mol

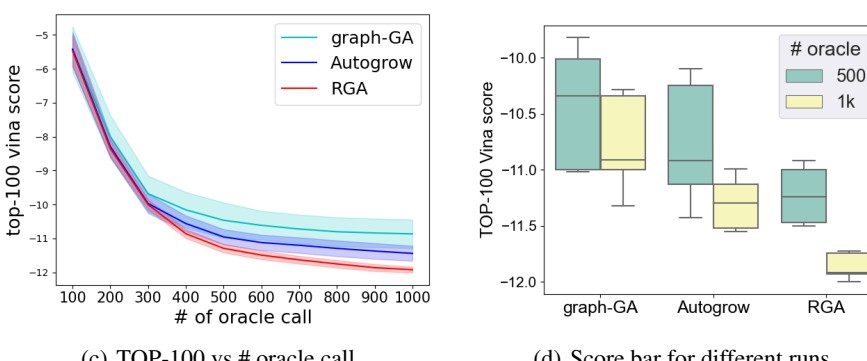

(c) TOP-100 vs # oracle call · · · · · (d) Score bar for different runs

Figure 2: (a) and (b): Example of ligand poses (generated by RGA) and binding sites of target structures. Example of 7l11: the PDB ID of target is 7l11, which is SARS-COV-2(2019-NCOV) main protease, the Vina score is -10.8 kcal/mol. Example of 3eml: the PDB ID is 3eml, which is a human A2A Adenosine receptor, the Vina score is -13.2 kcal/mol. (c) and (d): studies of suppressed random-walk behavior. (c) reports TOP-100 docking score as a function of oracle calls. The results are the means and standard deviations of 5 independent runs. (d) shows the bars of TOP-100 docking score for various independent runs.

Especially in the later learning phase (after 500 oracle calls), the policy networks are fine-tuned and guide the search more intelligently. This advantage leads to improved worst-case performance and a higher probability of successfully identifying bioactive drug candidates with constrained resources.

**Knowledge Transfer Between Protein Targets**. To verify if RGA benefited from learning the shared physics of ligand-target interaction, we conducted an ablation study whose results are in the last two rows of Table 1. Specifically, we compare RGA with two variants: (1) RGA-pretrain that does not pretrain the policy network with all native complex structures in the CrossDocked2020; (2) RGA-KT (knowledge transfer) that fine-tune the networks with data of individual target independently. We find that both strategies positively contribute to RGA on TOP-100/10/1 docking score. These results demonstrate the policy networks successfully learn the shared physics of ligand-target interactions and leverage the knowledge to improve their performance.

## 5   Conclusion

In this paper, we propose Reinforced Genetic Algorithm (RGA) to tackle the structure-based drug design problem. RGA reformulate the evolutionary process in genetic algorithms as a Markov decision process called evolutionary Markov decision process (EMDP) so that the searching processes could benefit from trained neural models. Specifically, we train policy networks to choose the parents to crossover and mutate instead of randomly sampling them. Further, we also leverage the common physics of the ligand-target interaction and adopt a knowledge-transfer strategy that uses data from other targets to train the networks. Through empirical study, we show that RGA has strong and robust optimization performance, consistently outperforming baseline methods in terms of docking score.

Though we adopted mutations originating from chemical reactions and the structural quality metrics seem good, we need to emphasize that the designed molecules from RGA do not guarantee synthesizability [49], as the crossover operations may break inheriting synthesizability. Directly working on

synthetic pathways could solve the problem [28, 59], but the extension is not trivial. As for future direction, we expect to theoretically analyze the EMDP formulation and the performance of RGA.

## Acknowledgments and Disclosure of Funding

T.F. and J.S. were supported by NSF award SCH-2205289, SCH-2014438, IIS-1838042, NIH award R01 1R01NS107291-01. W.G. and C.C. were supported by the Office of Naval Research under grant number N00014-21-1-2195 and the Machine Learning for Pharmaceutical Discovery and Synthesis consortium. Any opinions, findings, and conclusions or recommendations expressed in this material are those of the author(s) and do not necessarily reflect the views of the Office of Naval Research.

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
