_{\text{crossover}}^{\text{parent 1}} \in \mathcal{S}^{(t)}$ is $p_{\text{crossover}}^{(1)}(X_{\text{crossover}}^{\text{parent 1}}|\mathcal{S}^{(t)}) = \dfrac{\exp\left(\text{MLP}(\mathbf{h}_{\mathcal{T}\& X_{\text{crossover}}^{\text{parent 1}}})\right)}{\sum_{X' \in \mathcal{S}^{(t)}} \exp\left(\text{MLP}(\mathbf{h}_{\mathcal{T}\& X'})\right)}$, where $\mathcal{T}$ and $X$ denotes target and ligand (including 3D pose), respectively, $\mathcal{T}\& X$ denotes target-ligand complex. (2) The second action is to select the second parent ligand conditioned on the first parent ligand selected in the first action. Specifically, for ligand in the remaining population set, we concatenate the ENN's embedding of the target, first parent ligand $X_{\text{crossover}}^{\text{parent 1}}$ and the second parent ligand $X_{\text{mutation}}^{\text{parent 2}}$, and feed it into an MLP to estimate a scalar as an unnormalized probability. The unnormalized probabilities for all the ligands in the remaining population set are normalized via softmax function, i.e., $p_{\text{crossover}}^{(2)}(X_{\text{crossover}}^{\text{parent 2}}|X_{\text{crossover}}^{\text{parent 1}}, \mathcal{S}^{(t)}) = \text{Softmax}\big\{\text{MLP}(\mathbf{h}_{\mathcal{T}} \oplus \mathbf{h}_{X_{\text{crossover}}^{\text{parent 1}}} \oplus \mathbf{h}_{X_{\text{crossover}}^{\text{parent 2}}}), \cdots, \big\}_{X_{\text{crossover}}^{\text{parent 2}} \in \

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

## Contents

# A  Mathematical Notation

For ease of exposition, we list the mathematical notations in Table 2. All the mathematical notations are divided into three parts: (1) notation for genetic algorithm (Section 3.1); (2) notation for equivariance neural networks (ENN) [19] (Section 3.3); (3) notations for policy network (Section 3.3).

Table 2: Mathematical Notations. All the mathematical notations are divided into three parts: (1) notation for genetic algorithm (Section 3.1); (2) notation for equivariance neural networks (ENN) [19] (Section 3.3); (3) notations for policy network (Section 3.3).

| Notations | Descriptions |
|---|---|
| $X$ | ligand (drug molecule, including 3D pose) |
| $\mathcal{T}$ | target (target protein related to the disease) |
| $\mathcal{S}^{(t)}$ | the state (population of molecule) at the $t$-th generation. |
| $\mathcal{Q}^{(t)}$ | offspring pool at the $t$-th generation. |
| $K$ | the number of molecule in the state, i.e., size of population. |
| $X_{\text{crossover}}^{\text{parent 1/2}}$ | the first/second parent molecule in the crossover. |
| $X_{\text{crossover}}^{\text{child 1/2}}$ | the first/second child molecule in the crossover. |
| $X_{\text{mutation}}^{\text{parent}}$ | parent molecule in the mutation |
| $X_{\text{mutation}}^{\text{child}}$ | child molecule in the mutation |
| $\xi \in \mathcal{R}$ | the selection reaction in the mutation |
| $\mathcal{R}$ | the reaction set (library) for mutation |
| ENN | equivariance neural networks [19] |
| $\mathcal{V} = \{H, C, O, N, \cdots\}$ | vocabulary set of atoms |
| $\mathcal{Y} = (\mathcal{A}, \mathcal{Z})$ | 3D structure |
| $\mathcal{A}$ | categories of all the atoms |
| $\mathbf{a}_i$ | one-hot vector that encode category of $i$-th atom |
| $\mathcal{Z}$ | 3D coordinates of the atoms |
| $\mathbf{D} \in \mathbb{R}^{|\mathcal{V}| \times d}$ | the embedding matrix of all the categories of atoms |
| $d$ | the hidden dimension in ENN. |
| $N$ | number of atoms in the input of ENN. |
| $L$ | number of layers in ENN |
| $l = 0, 1, \cdots, L$ | index of layer in ENN |
| MLP | multiple layer perceptrons |
| $\text{MLP}_e(\cdot), \text{MLP}_x(\cdot), \text{MLP}_h(\cdot)$ | two-layer MLP in ENN with Swish activation [51] in hidden layer |
| $\oplus$ | the concatenation of vectors |
| $\mathbf{Z}^{(0)} = \{\mathbf{z}_i\}_{i=1}^N$ | initial coordinate embeddings, real 3D coordinates of all the nodes. |
| $\mathbf{H}^{(l)} = \{\mathbf{h}_i^{(l)}\}_{i=1}^N$ | Node embeddings at the $l$-th layer |
| $\mathbf{h}_i^{(0)} = \mathbf{D}^\top \mathbf{a}_i \in \mathbb{R}^d$ | The initial node embedding that embeds the $i$-th node |
| $\mathbf{Z}^{(l)} = \{\mathbf{z}_i^{(l)}\}_{i=1}^N$ | Coordinate embeddings at the $l$-th layer |
| $\mathbf{w}_{ij}^{(l)}$ | message vector for the edge from node $i$ to node $j$ at $l$-th layer |
| $\mathbf{v}_i^{(l)}$ | message vector for node $i$ at $l$-th layer |
| $\mathbf{z}_i^{(l)}$ | the position embedding for node $i$ at $l$-th layer |
| $\mathbf{h}_i^{(l)}$ | the node embedding for node $i$ at $l$-th layer |
| $\mathbf{h}_{\mathcal{Y}} = \text{ENN}(\mathcal{Y})$ | ENN representation of the 3D graph $\mathcal{Y}$ (Equation 1) |
| $p_{\text{crossover}}^{(1)}(X_{\text{crossover}}^{\text{parent 1}}|\mathcal{S}^{(t)})$ | probability to select the first parent molecule in crossover |
| $p_{\text{crossover}}^{(2)}(X_{\text{crossover}}^{\text{parent 2}}|X_{\text{crossover}}^{\text{parent 1}}, \mathcal{S}^{(t)})$ | probability to select the second parent molecule in crossover |
| $p_{\text{crossover}}(X_{\text{crossover}}^{\text{child 1}}, X_{\text{crossover}}^{\text{child 2}}|\mathcal{S}^{(t)})$ | probability of two generated child molecules in crossover (Eq 2) |
| $p_{\text{mutation}}^{(1)}(X_{\text{mutation}}^{\text{parent}}|\mathcal{S}^{(t)})$ | probability to select the parent molecule in mutation |
| $p_{\text{mutation}}^{(2)}(\xi|X_{\text{mutation}}^{\text{parent}}, \mathcal{S}^{(t)})$ | probability to select the reaction in mutation |
| $p_{\text{mutation}}(X_{\text{mutation}}^{\text{child}}|\mathcal{S}^{(t)})$ | probability of generated child molecule in mutation (Eq 3) |

## B   Illustration of Genetic Algorithm

Figure 3 provides two examples to illustrate crossover and mutation operations in genetic algorithm described in Section 3.1.

**Crossover**, also called recombination, combines the structure of two parents to generate new children. Following Autogrow 4.0 [17], as shown in Figure 3(a), we select two parents from the last generation and search for the largest common substructure shared between them. Then we generate two children by randomly switching their decorating moieties, i.e., the side chains attached to the common substructure.

**Mutation** operator performs an *in silico* chemical reaction to generate an altered child compound (i.e., product in the chemical reaction) derived from a parent (reactant in the chemical reaction), as shown in Figure 3(b). The chemical reaction here contains two reactants, one is parent molecule, another is from reaction set. The reaction set $\mathcal{R}$ is generated via merging two public reaction libraries: (1) the AutoClickChemRxn set (36 reactions) [47] and (2) RobustRxn set (58 reactions [50]). Each reaction $\xi$ in reaction set $\mathcal{R}$ contains a SMARTS string based reaction template and a reactant. It uses SMARTS reaction template, together with RDKit [60], to perform chemical mutations efficiently. The process is written as $X^{\text{parent}}_{\text{mutation}} \xrightarrow{\text{mutated by } \xi} X^{\text{child}}_{\text{mutation}}$, where $\xi$ is selected reaction (with a reaction template and another reactant). The ligand to be mutated and the reaction used for mutation are both randomly selected from previous generation and reaction set $\mathcal{R}$, respectively. Compared with the mutation operator in conventional GA that randomly flipping an arbitrary bit, the reaction-based mutation enhance synthesizability of the generated molecules [17]. Mutation operator performs an *in silico* chemical reaction to generate an altered child compound (i.e., product in the chemical reaction) derived from a parent (reactant in the chemical reaction). The chemical reaction here contains two reactants, one is parent molecule, another is from reaction set.

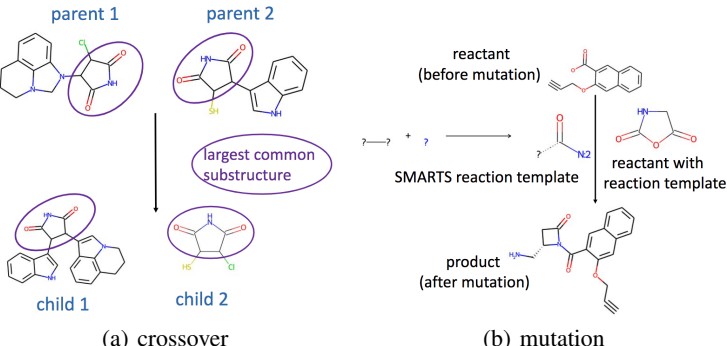

(a)  crossover               (b)  mutation

Figure 3: Illustration of GA operations: (a) **crossover** finds the largest substructure that the two parent compounds share and generates a child by combining their decorating moieties. (b) **mutation**: given a reactant (i.e., parent), mutation operator uses SMARTS-reaction template (with another reactant) to performs an *in silico* chemical reaction to generate child compound (i.e., product).

## C   Baseline Setup

In this section, we describe the experimental setting for baseline methods. Most of the settings follow the original papers.

- **GA+D** (genetic algorithm enhanced by discriminator neural network) [27] utilizes SELFIES string as the representation of molecules, thus guaranteeing the 100% chemical validity of the generated molecules. Following their original paper, the discriminator neural network is a two-layer fully connected neural network with ReLU activation and sigmoid output layer. The hidden size is 100. the size of the output layer is 1. The input feature of discriminator neural network is a vector of chemical and geometrical properties characterizing the molecules. The population size is set to 300. Maximal generation number is set to 1000. The patience is set to 5. When the property does not improve when the patience exhausts, the process early stops. We used Adam optimizer with 1e-3 as the initial learning rate. beta

($\beta$) is the weight of discriminator neural network's score in fitness evaluation, which is used to select most promising molecules in each generation. We set $\beta = 1$.

- **Graph-GA** [16] uses molecular graph to represent molecules and uses crossover and mutation operations to edit the molecular graph. After tuning, the size of population is set to 120. The size of offspring is set to 70. The mutation rate is set to 0.067. Graph-GA do not have learnable parameters and is easy to implement. We use the implementation in GuacaMol [13].

- **MolDQN** (Molecule Deep Q-Networks) [31] uses molecular graph to represent molecules, formulate the molecule optimization process as a Markov Decision Process (MDP) and utilize Deep Q-value learning to optimize it. It grows molecular graph atom-wise, that is, in each episode, it adds one atom to the partially generated molecular graph. The reward function is the negative value of the Vina score. Following the original paper, maximal step in each episode is 40. Each step calls oracle once. The discount factor is 0.9. Deep Q-network is a multilayer perceptron (MLP) whose hidden dimensions are 1024, 512, 128, 32, respectively. The model size is 6.4 M. The input of the Q-network is the concatenation of the molecule feature (2048-bit Morgan fingerprint, with a radius of 3) and the number of left steps. Adam is used as an optimizer with 1e-4 as the initial learning rate. Only rings with a size of 5 and 6 are allowed. It leverages $\epsilon$-greedy together with randomized value functions (bootstrapped-DQN) as an exploration policy, $\epsilon$ is annealed from 1 to 0.01 in a piecewise linear way.

- **RationaleRL** [32]. The architecture of the generator is a message-passing network (MPN) followed by MLPs applied in breadth-first order. The generator is pre-trained on general molecules combined with an encoder and then fine-tuned to maximize the reward function using policy gradient. The encoder and decoder MPNs both have hidden dimensions of 400. The dimension of the latent variable is 50. Adam optimizer is used on both pre-training and fine-tuning with initial learning rates of 1e-3, 5e-4, respectively. The annealing rate is 0.9. We pre-trained the model with 20 epochs.

- **MARS** [36] leverage Markov chain Monte Carlo sampling (MCMC) on molecules with an annealing scheme and an adaptive proposal. The proposal is parameterized by a graph neural network, which is trained on MCMC samples. We follow most of the settings in the original paper. The message passing network has six layers, where the node embedding size is set to 64. Adam is used as an optimizer with 3e-4 initial learning rate. To generate a basic unit, top-1000 frequent fragments are drawn from ZINC database [56] by enumerating single bonds to break. During the annealing process, the temperature $T = 0.95^{\lfloor t/5 \rfloor}$ would gradually decrease to 0.

- **Autogrow 4.0** [17] is the base model for RGA and have been briefly described in Section 3.1. The setup of Autogrow is the same as RGA for fair comparison, the only difference is that RGA use policy network to guide the selection of ligands and reaction for crossover and mutation while Autogrow randomly selects them. The reaction set $\mathcal{R}$ is generated via merging two public reaction libraries: (1) the AutoClickChemRxn set (36 reactions) [47] and (2) RobustRxn set (58 reactions [50]). In each generation, It generates 200 offspring (100 from crossover and 100 from mutation) and keep 50 most promising (with lowest Vina scores) ones for the next generation.

- **Screening** exhaustively searches the ZINC database [56] within oracle budget. It is the traditional high-throughput screening approach.

- **REINVENT** [29] is a reinforcement learning approach, represent molecule as SMILES string and uses recurrent neural network to model SMILES string. It pretrains a prior model using molecules on ZINC and finetune the model using the reward function. It uses REINFORCE to maximize the expected reward function. The learning rate is set to 0.0005; the batch size is set to 64; The hyperparameter $\sigma$ weighs the pretrained prior model and the reward function, and is set to 60. The model size is 16.3M.

- **JTVAE** [21] build a junction tree to represent molecule via using substructure (either ring or atom) to represent molecule. It uses both molecular graph-leven and junction tree-level encoder and decoder. The VAE model is pretrained on ZINC databases. Then Bayesian Optimization is used to optimize the docking score on the continuous latent space. We use "botorch", the python's Bayesian optimization package, to implement the Bayesian

optimization process. It has 703 substructures in vocabulary, extracted from ZINC. The hidden size is 450. The latent size of VAE is set to 56. The model size is 21.8 M.

- **Gen3D** [10] uses 3D deep generative models and grow the molecule via adding atoms auto-regressively. It train a universal model for all the targets. The number of message passing layers in context encoder is 6, and the hidden dimension is 256. We train the model using the Adam optimizer at learning rate 0.0001. The model size is 17.4 M.

- **GEGL** (Genetic Expert Guided Learning) [33] uses LSTM (guided by RL agent) to imitate GA process, however, it is unable to inherit the GA's flexible assembling manner due to the auto-regressive essence of LSTM. It use Adam as optimizer with initial learning rate 1e-3. The batch size during sampling is 512, the batch size during optimization is 256. In GA, mutation rate is 0.01. The similarity threshold is 0.4, which constrain the similarity between the original molecule and the edited molecule. The maximal SMILES length is set to 120.

## D  Additional Experimental Setup

### D.1  Docking Simulation

Molecular docking is a computational method which predicts the preferred orientation of one molecule to a second when a ligand and a target are bound to each other to form a stable complex. Knowledge of the preferred orientation in turn may be used to predict the strength of association or binding affinity between two molecules using, for example, scoring functions. We adopt AutoDock Vina [52] to evaluate the binding affinity. The docking score estimated by AutoDock Vina is called Vina score and roughly characterizes the free energy changes of binding processes in kcal/mol. Vina score is usually smaller than 0 and lower Vina score means a stronger binding affinity between the ligand and target. We leverage the negative value of the docking score as reward function (Equation 4).

### D.2  Dataset

In this paper, we use ZINC [56] database and CrossDocked2020 [57] dataset. ZINC is a free database of 250 thousands commercially-available drug-like chemical compounds for virtual screening [56]. We randomly select molecules from ZINC [56] database (around 250 thousands drug-like molecules) as 0-th generation of the genetic algorithms (`RGA`, Autogrow 4.0, graph-GA GA+D). Other baseline methods also use ZINC to either pretrain the models, e.g., JTVAE, REINVENT, RationaleRL or provide searching database, e.g., screening. We adopt CrossDocked2020 [57] dataset that contains around 22 million ligand-protein complexes as the training data for pretraining the policy neural networks, as mentioned in Section 3.3.

Regarding the target proteins, we picked various disease-related proteins, including G-protein coupling receptors (GPCRs) and kinases from DUD-E [53] and the SARS-CoV-2 main protease [54] as targets. for all the selected target protein, the binding pocket size for all the targets are set to (15.0, 15.0, 15.0). The units of coordinate are Angstrom $\mathring{A}$ ($10^{-10}$ m). Detailed descriptions of these targets are available at `https://www.rcsb.org/`.

### D.3  Evaluation metrics

We leverage the following evaluation metrics to measure the optimization performance:

- **Novelty** is the fraction of the generated molecules that do not appear in the training set.
- **Diversity** of generated molecules is defined as the average pairwise Tanimoto distance between the Morgan fingerprints [30, 32, 36].

$$\text{diversity} = 1 - \frac{1}{|\mathcal{M}|(|\mathcal{M}| - 1)} \sum_{m_1, m_2 \in \mathcal{M}} \text{sim}(m_1, m_2), \qquad (5)$$

  where $\mathcal{M}$ is the set of generated molecules that we want to evaluate. $\text{sim}(m_1, m_2)$ is the Tanimoto similarity between molecule $m_1$ and $m_2$, where (Tanimoto) Similarity measures the similarity between the input molecule and generated molecules. It is defined as $\text{sim}(X, Y) = \frac{\text{FP}_X^\top \text{FP}_Y}{\|\text{FP}_X\|_2 \|\text{FP}_Y\|_2}$, $\text{FP}_X$ is the binary Morgan fingerprint vector for the molecule $X$. In this paper, it is a 2048-bit binary vector.

- **QED** represents a quantitative estimate of drug-likeness. QED score ranges from 0 to 1. It can be evaluated by the RDKit package (`https://www.rdkit.org/`).

- **SA** (Synthetic Accessibility) score measures how hard it is to synthesize a given molecule, based on a combination of the molecule's fragments contributions [61]. It is evaluated via RDKit [60]. The raw SA score ranges from 1 to 10. A higher SA score means the molecule is hard to be synthesized and is not desirable.

- **Run Time**. Unlike optimizing some simple oracles such as QED and LogP scores, the docking simulation need to search 3D molecular conformation docked in the target, which is computationally expensive. Thus run time is an important metric to measure the efficiency of the methods.

## E  Implementation Details

### E.1  Software/Hardware Configuration

We implemented `RGA` using Pytorch 1.10.2, Python 3.7, RDKit v2020.09.1.0 on an Intel Xeon E5-2690 machine with 256G RAM and NVIDIA Pascal Titan X GPUs.

### E.2  Hyperparameter Setup

The neural architectures of policy networks are E(3)-equivariant neural network (ENN) [19]. The vocabulary set $\mathcal{V} = \{C, N, O, S, H, \text{other}\}$. In ENN, the number of layers is set to 3, i.e., $L = 3$; the hidden dimension is set to 100, i.e., $d = 100$. In Equation 1, $\text{MLP}_e(\cdot), \text{MLP}_x(\cdot), \text{MLP}_h(\cdot)$ are two-layer MLP in ENN with Swish activation [51] in hidden layer. Summation function is used as aggregation function to aggregate the last-layer's node embedding into graph-level embedding. All the atoms that within the binding site are used as the input of ENN. REINFORCE is used to implement policy gradient [62, 29]. Adam is utilized as optimizer with learning rate 0.001 for both crossover and mutation policy networks. The reaction set $\mathcal{R}$ is generated via merging two public reaction libraries: (1) the AutoClickChemRxn set (36 reactions) [47] and (2) RobustRxn set (58 reactions [50]). We use RDKit [60] to perform *in silico* chemical reaction based on SMARTS reaction template, then we have $|\mathcal{R}| = 36 + 58 = 94$. In each generation, we generate up to 200 offspring (100 from crossover and 100 from mutation) and keep 50 most promising (with lowest Vina scores) ones for the next generation, i.e., $K = 50$.

### E.3  Code Repository

The code repository is uploaded in supplementary material for reproducibility.

## F  Additional Experiment

### F.1  Efficiency Study

As mentioned before, unlike optimizing some simple oracles such as QED and LogP scores, the docking simulation need to search 3D molecular conformation docked in the target, which is time-consuming. Thus run time is an important measurement to evaluate the efficiency of the methods. We report the bar of run time over different targets for all the compared methods in Figure 4. The run times varies greatly over different methods. Thus, for ease of visualization, we divided all the methods into two groups. One is slow group, containing 5 methods: screening, GEGL, REINVENT, RationaleRL and JTVAE, where all the methods take more than 10 hours. Another is fast group (<10 hours), containing 7 methods, MARS, MolDQN, Gen3D, GA+D, GraphGA, Autogrow, and RGA. We find that both Autogrow and RGA are efficient compared with other methods. This attributes to the unique design of genetic algorithm (both crossover and mutation operations) and the usage of filter after GA operators, as described in Section 3.1. RGA is only slightly slower than Autogrow because it requires additional computation to pretrain/train the policy neural networks.

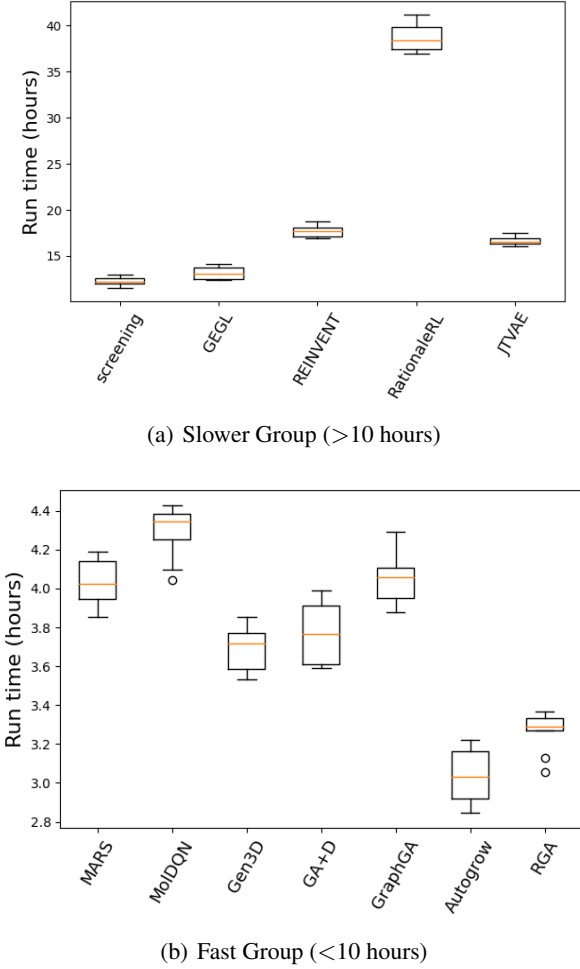

(a) Slower Group (>10 hours)

(b) Fast Group (<10 hours)

Figure 4: Efficiency evaluation measured by run time for all the methods. The unit of run time is hours. Due to the big variance in run time, for ease of visualization, we divide all the methods into two groups. One is slow group, containing 5 methods: screening, GEGL, REINVENT, RationaleRL and JTVAE. Another is fast group, containing 7 methods, MARS, MolDQN, Gen3D, GA+D, GraphGA, Autogrow, and RGA.

## F.2   Pretraining Equivariant neural network

Pretraining equivariant neural network is a crucial step to RGA. We adopt CrossDocked2020 [57] dataset that contains around 22 million ligand-protein complexes as the training data for pretraining the policy neural networks, as mentioned in Section 3.3. We split the whole dataset into training/validation dataset with ratio of 9:1. Each data point is a target-ligand complex and the binding affinity (scalar). We report the validation loss of ENN on target-ligand binding affinity prediction task. The validation loss function is root mean square error (RMSE). We find the learning process converges rapidly when passing 150K data points (within an epoch) in terms of validation RMSE loss.

## F.3   Additional Ablation Study

To further understand our model and GA process, we conduct an ablation study to investigate the impact of each component/strategy to optimization performance. Specifically, we consider the following four variants of RGA. RGA-pretrain is a variant of RGA that does not pretrain the policy neural network. RGA-KT (Knowledge Transfer) is a variant of RGA that does not training policy neural network on different target proteins, i.e., optimizing ligand for one target at a time. RGA-MU (mutation) is a variant of RGA that does not involve mutation operation in GA. That is, all the ligands

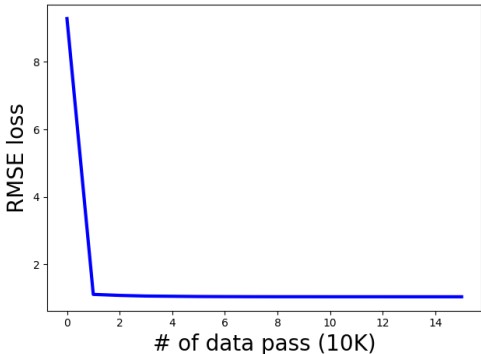

Figure 5: Learning curve of pretraining Equivariant neural network based on target-ligand binding affinity prediction. We plot the root-mean-square error (RMSE) loss as a function of number of passed data. We use early stop strategy to terminate the learning process earlier when the validation loss would not decrease to save computational resource and avoid overfitting. We found the learning process would converges when passing 150K data samples (within an epoch), RMSE loss decreases from more than 8 to less than 1.

Table 3: Ablation studies. Arrows ($\uparrow$, $\downarrow$) indicate the direction of better performance. For each metric, the best method is underlined. RGA-pretrain is a variant of RGA that does not pretrain the policy neural network. RGA-KT (Knowledge Transfer) is a variant of RGA that does not training policy neural network on different target proteins, i.e., optimizing ligand for one target at a time. RGA-MU (mutation) is a variant of RGA that does not involve mutation operation in GA. That is, all the ligands are generated via crossover operator. Correspondingly, RGA-CO (crossover) is a variant that does not use crossover operation in GA, which means no mutation operator. Via comparing the results with RGA (full) in the first line, we observe that removing either component would cause a drop in optimization performance (i.e., increase in TOP-100/10/1 scores).

| Method | TOP-100$\downarrow$ | TOP-10$\downarrow$ | TOP-1$\downarrow$ | Nov$\uparrow$ | Div$\uparrow$ | QED$\uparrow$ | SA$\downarrow$ |
|---|---|---|---|---|---|---|---|
| RGA (full) | $-11.867_{\pm0.170}$ | $-12.564_{\pm0.287}$ | $-12.869_{\pm0.473}$ | $100.0_{\pm0.0\%}$ | $0.857_{\pm0.020}$ | $0.742_{\pm0.036}$ | $2.473_{\pm0.048}$ |
| RGA - pretrain | $-11.443_{\pm0.219}$ | $-12.424_{\pm0.386}$ | $-12.435_{\pm0.654}$ | $100.0_{\pm0.0\%}$ | $0.854_{\pm0.035}$ | $0.750_{\pm0.034}$ | $2.494_{\pm0.043}$ |
| RGA - KT | $-11.434_{\pm0.169}$ | $-12.437_{\pm0.354}$ | $-12.502_{\pm0.603}$ | $100.0_{\pm0.0\%}$ | $0.853_{\pm0.028}$ | $0.738_{\pm0.034}$ | $2.501_{\pm0.050}$ |
| RGA - MU | $-10.919_{\pm0.166}$ | $-11.135_{\pm0.362}$ | $-11.747_{\pm0.455}$ | $100.0_{\pm0.0\%}$ | $0.812_{\pm0.032}$ | $0.702_{\pm0.050}$ | $2.970_{\pm0.048}$ |
| RGA - CO | $-9.866_{\pm0.169}$ | $-10.320_{\pm0.296}$ | $-10.793_{\pm0.501}$ | $100.0_{\pm0.0\%}$ | $0.737_{\pm0.048}$ | $0.748_{\pm0.067}$ | $2.467_{\pm0.034}$ |

are generated via crossover operator. Correspondingly, RGA-CO (crossover) is a variant that does not use crossover operation in GA, which means no mutation operator. The results are reported in Table 3. We find that removing either component/strategy will cause a drop in optimization performance (i.e., increase in TOP-100/10/1 scores). Both crossover and mutation are critical to the optimization performance. Also, both pretraining the policy networks and knowledge transfer between different target have positive contribution to the performance. The ablation study furtherly validates the effectiveness of the proposed RGA method.

## F.4 Scalability

As mentioned before, the population size in RGA is $K$. Then we analyze the computational complexity within each generation. There are two operations, including crossover and mutation operations, as described in Section 3.3.

For crossover, the first step is to select the first parent molecule, we need to evaluate the probability over all the molecules in current population, whose complexity is $O(K)$. The second step is to select the second parent molecule based on the first parent molecule, we need to evaluate the probability over all the remaining molecules in current population, as shown in Equation (2), whose complexity is $O(K-1)$. The complexity of crossover operation is $O(K)$.

On the other hand, for mutation operation, the first step is to select the parent molecule (only one parent) via evaluating the probability over all the molecules in the current population, i.e., $p_{\text{mutation}}^{(1)}(X_{\text{mutation}}^{\text{parent}}|\mathcal{S}^{(t)})$, whose complexity is $O(K)$. The second step is to select a mutated molecules

Table 4: Results of hypothesis testing. We conduct hypothesis testing to show the statistical significance of our method over other GA-based methods. Specifically, we compare the significance of the improvement over GA methods (AutoGrow4, Graph-GA, GA+D, GEGL) via running 5 independent trials with different random seeds and then evaluating p-value. We consider top-100/10/1 scores, the most important metrics for optimization performance.

|  | p-value on TOP-100 | p-value on TOP-10 | p-value on TOP-1 |
| --- | --- | --- | --- |
| RGA v.s. AutoGrow 4 | 0.002 | 0.005 | 0.07 |
| RGA v.s. Graph-GA | 0.003 | 0.010 | 0.046 |
| RGA v.s. GA+D | 1.0e-7 | 5.0e-5 | 3e-4 |
| RGA v.s. GEGL | 2.5e-4 | 3.7e-3 | 5.0e-3 |

(generated by chemical reaction), as shown in Equation (3). The complexity is $O(|\mathcal{R}|)$, where $\mathcal{R}$ is the reaction set (see Table 2). The complexity of mutation operation is $O(|\mathcal{R}| + K)$.

To summarize, the complexity of RGA is $O(K + |\mathcal{R}|)$. That is, RGA scales linearly in population size $K$ and the size of the chemical reaction set $|\mathcal{R}|$. As mentioned in Section E.2, $|\mathcal{R}| = 94$, $K = 50$. Thus, RGA owns desired scalability and is not computational expensive.

### F.5 Significance Studies

In this section, we present the significance studies. Specifically, we conduct the hypothesis testing to show the statistical significance of our method over the other GA methods. We compare the significance of the improvement over other GA methods (including AutoGrow4, Graph-GA, GA+D, and GEGL) via running 5 independent trials with different random seeds and then evaluating the **p-value**. We consider the top-100 and top-10 score as the major metrics, which are the most important metrics for optimization performance. We compare RGA versus AutoGrow 4, Graph-GA and GA+D. The results (p-value) are shown in Table 4. We find almost all the p-values are less than 0.05 (except one value), which indicates that the improvements of RGA over other GA methods are statistically significant.

### F.6 Example of the generated ligand

This section shows some examples of the generated ligands with desirable binding affinity for various target proteins in Figure 6, 7, 8 and 9, respectively. We observe that the generated ligands bind tightly with the target proteins.

## G Additional Discussion on Related Work

**Methodology**. The molecule generations methods can be divided into two categories. The first one is deep generative models (DGMs), which leverage the continuous representation to estimate the data distribution using various kinds of deep neural networks, including variational autoencoder (VAE) [20, 21], generative adversarial network (GAN) [22, 23], normalizing flow model [24, 63, 25], energy based model [26] and diffusion model [64], etc. The second one is combinatorial optimization methods, which directly search the discrete chemical space, including genetic algorithm (GA) [16, 27, 28], reinforcement learning approaches (RL) [29, 30, 31, 32], Bayesian Optimization (BO) [34, 65], Monte Carlo Tree Search (MCTS) [66, 67, 32] and Markov Chain Monte Carlo (MCMC) [35, 36, 37]. Deep generative models (DGMs) usually require a large amount of data to fit, which impedes their usage in low data regime. In contrast, combinatorial optimization methods require less training data, while the trade-off is the need to call the optimization oracles during the exploration in the chemical space [31, 38, 49, 28, 13].

Among all the machine learning methods, Genetic algorithm (GA) exhibits superior performance in some standard benchmarks [13, 14, 15]. The key reason is GA's global assembling strategy. Specifically, in each generation (iteration), GA maintains a population of molecule candidates (a.k.a. parents), and conducts the crossover operation between two (random-selected) parent candidates, which enables relatively large exchanges on molecular sub-graph between molecular graphs. However,

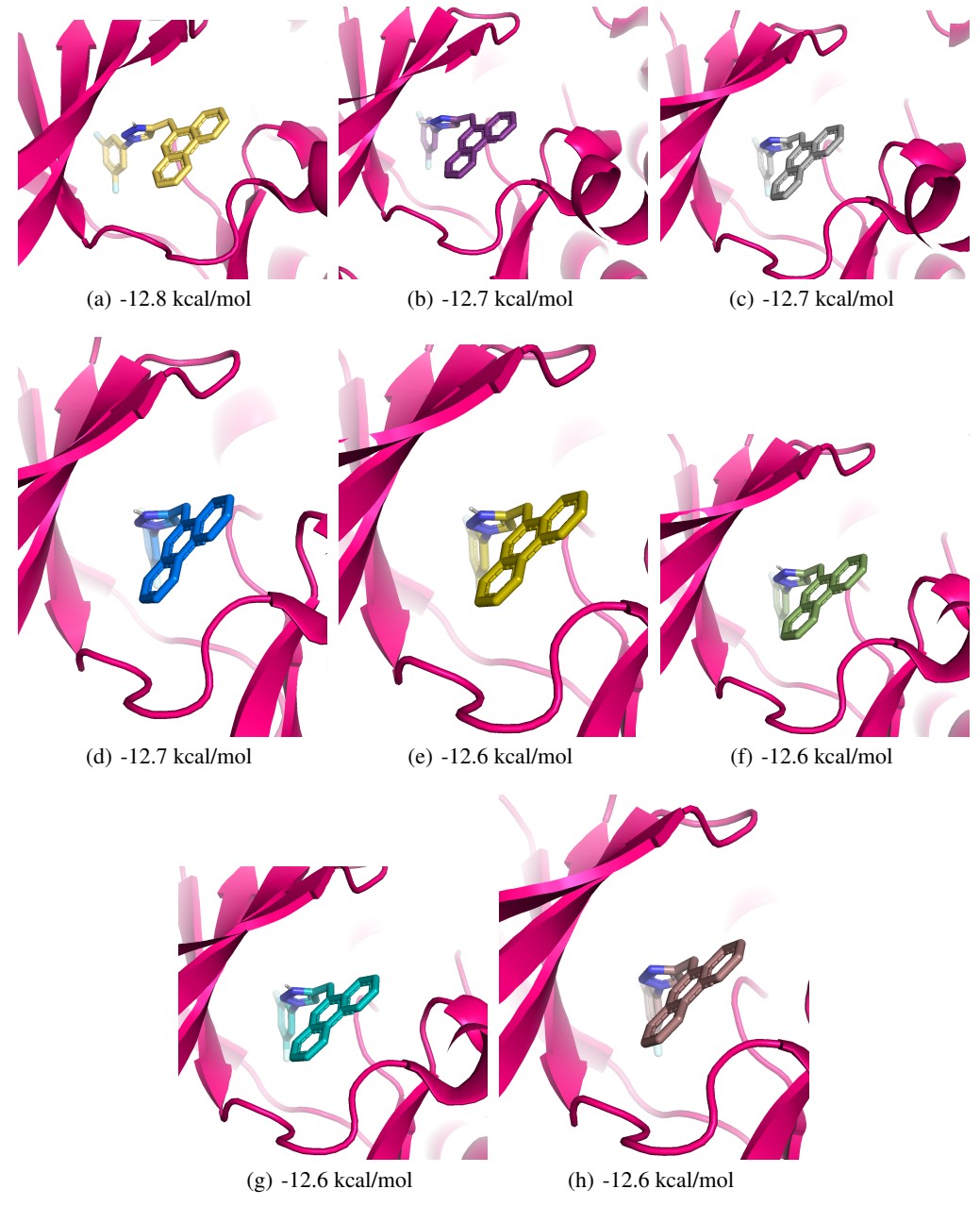

(a) -12.8 kcal/mol     (b) -12.7 kcal/mol     (c) -12.7 kcal/mol

(d) -12.7 kcal/mol     (e) -12.6 kcal/mol     (f) -12.6 kcal/mol

(g) -12.6 kcal/mol     (h) -12.6 kcal/mol

Figure 6: Example of ligand poses (generated by `RGA`) and binding sites of target structures "2rgp".

GA is leveraging random-walk based mutation and crossover operations [16, 17] and is essentially based on brute-force trial and error.

On the other hand, Reinforcement learning (RL) methods are good at navigating the discrete space via prioritizing the promising searching branches and circumventing brute-force search. The current RL-based methods [29, 30, 31, 32] slightly left behind other state-of-the-art combinatorial optimization methods [38, 14]. The main reason is that current RL based molecule optimization approaches are based on auto-regressive assembling strategy, i.e., growing molecules iteratively via adding a basic building block one time, where the building block can be either a token in SMILES representation [29] or a substructure in molecular graph representation [30, 31, 32]. Such assembling strategy are essentially local search methods, which hinders the algorithm's ability to overcome the rough optimization landscape (or energy barrier) and is easy to be stuck in the local optimum [68, 69].

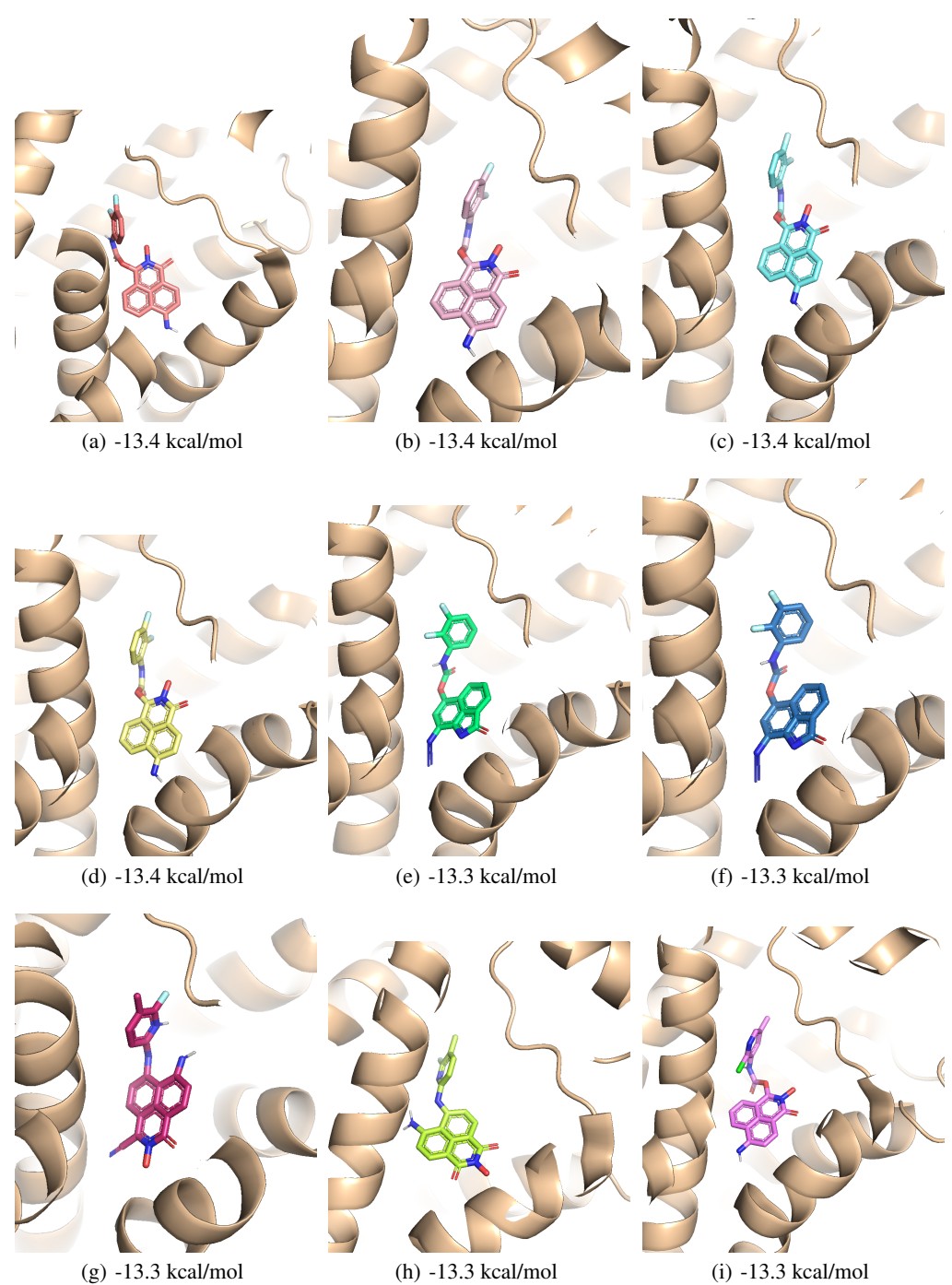

(a) -13.4 kcal/mol     (b) -13.4 kcal/mol     (c) -13.4 kcal/mol

(d) -13.4 kcal/mol     (e) -13.3 kcal/mol     (f) -13.3 kcal/mol

(g) -13.3 kcal/mol     (h) -13.3 kcal/mol     (i) -13.3 kcal/mol

Figure 7: Example of ligand poses (generated by RGA) and binding sites of target structures "3ny8".

**Discussion**. Among all the machine learning methods, molecular graph level genetic algorithm (GA) exhibits state-of-the-art performance in some standard molecule optimization benchmarks [13, 14, 15]. The key reason is GA's assembling manner. Specifically, in each generation (iteration), GA maintains a population of possible candidates (a.k.a. parents), and conducts the crossover operation between two candidates to generate new offspring, which enables thorough exploration to the chemical space. However, there is still improvement space for GA. GA are leveraging random-walk based mutation and crossover operations [16] and suffers from brute-force trial and error strategy.

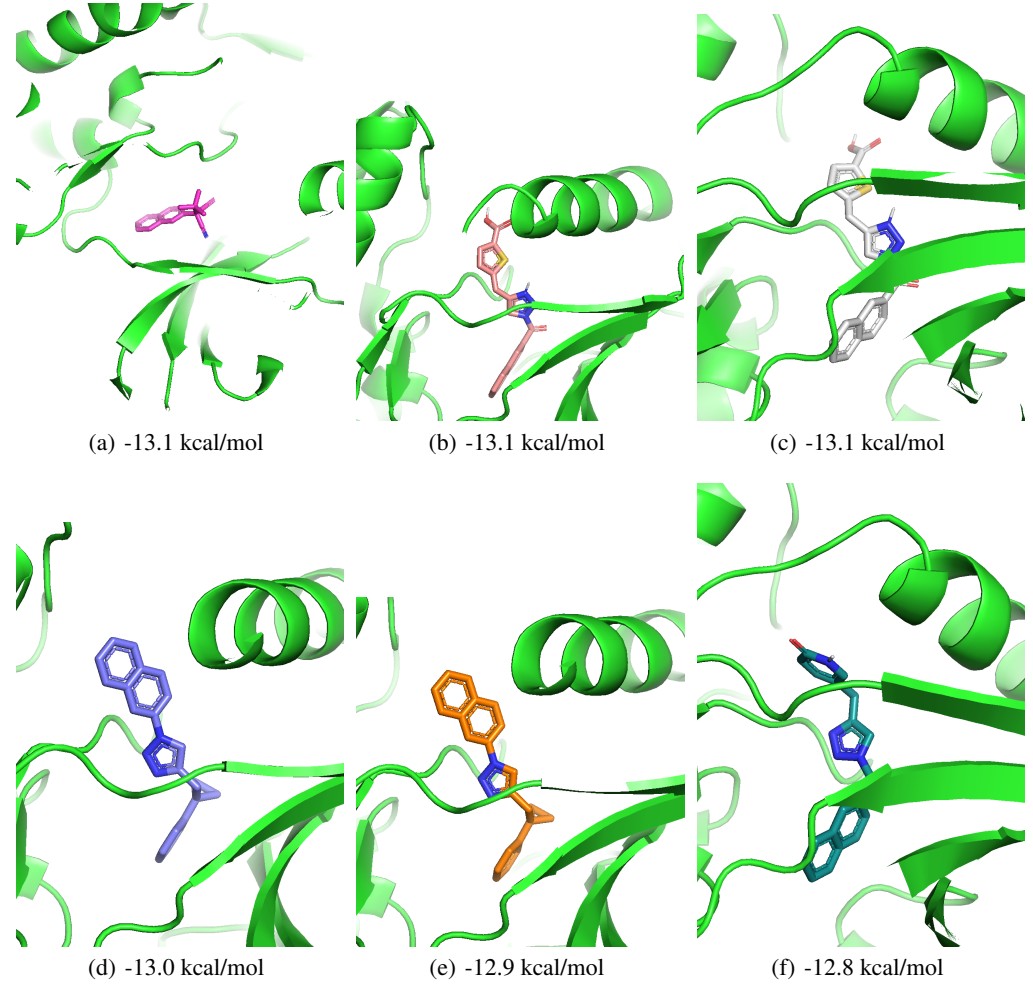

| (a) -13.1 kcal/mol | (b) -13.1 kcal/mol | (c) -13.1 kcal/mol |
| (d) -13.0 kcal/mol | (e) -12.9 kcal/mol | (f) -12.8 kcal/mol |

Figure 8: Example of ligand poses (generated by `RGA`) and binding sites of target structures "1iep".

On the other hand, reinforcement learning approaches are good at navigating the discrete space via prioritizing the promising decisions that are worth investigating, for example, AlphaGo successfully applied RL to defeat a professional human Go player [70]. However, the current RL based drug design methods [29, 30, 31] slightly left behind other state-of-the-art combinatorial optimization methods. The main reason lies at the inferior assembling strategy, which grows molecule in an auto-regressive fashion. It is hard for this kind of local search strategy to overcome the barrier of the objective, so it is easy to be trapped into the local optimum.

Deep learning methods can also enhance genetic algorithm. Due to the random selection used in genetic algorithm, it is challenging to apply deep learning methods in the generation of new candidates (molecules in this paper). Deep learning can be used to compose fitness evaluation to select the offspring. For example, GA+D [27] leverages deep neural network as a discriminator to measure the drug's proximity to the training data, which is incorporated as a scorer in fitness evaluation. [71] train a deep neural network-based property predictor and leverage it to enhance the evolutionary algorithm.

In this paper, we attempt to enhance genetic algorithm using reinforcement learning technique. Specifically, we propose Reinforced Genetic Algorithm (`RGA`), which inherits the assembling manner from genetic algorithm and use reinforcement learning to guide the search over the chemical space. [33] also combine RL and GA, which uses LSTM (guided by RL agent) to imitate GA process, however, it is unable to inherit the GA's flexible assembling manner due to the auto-regressive essence of LSTM.

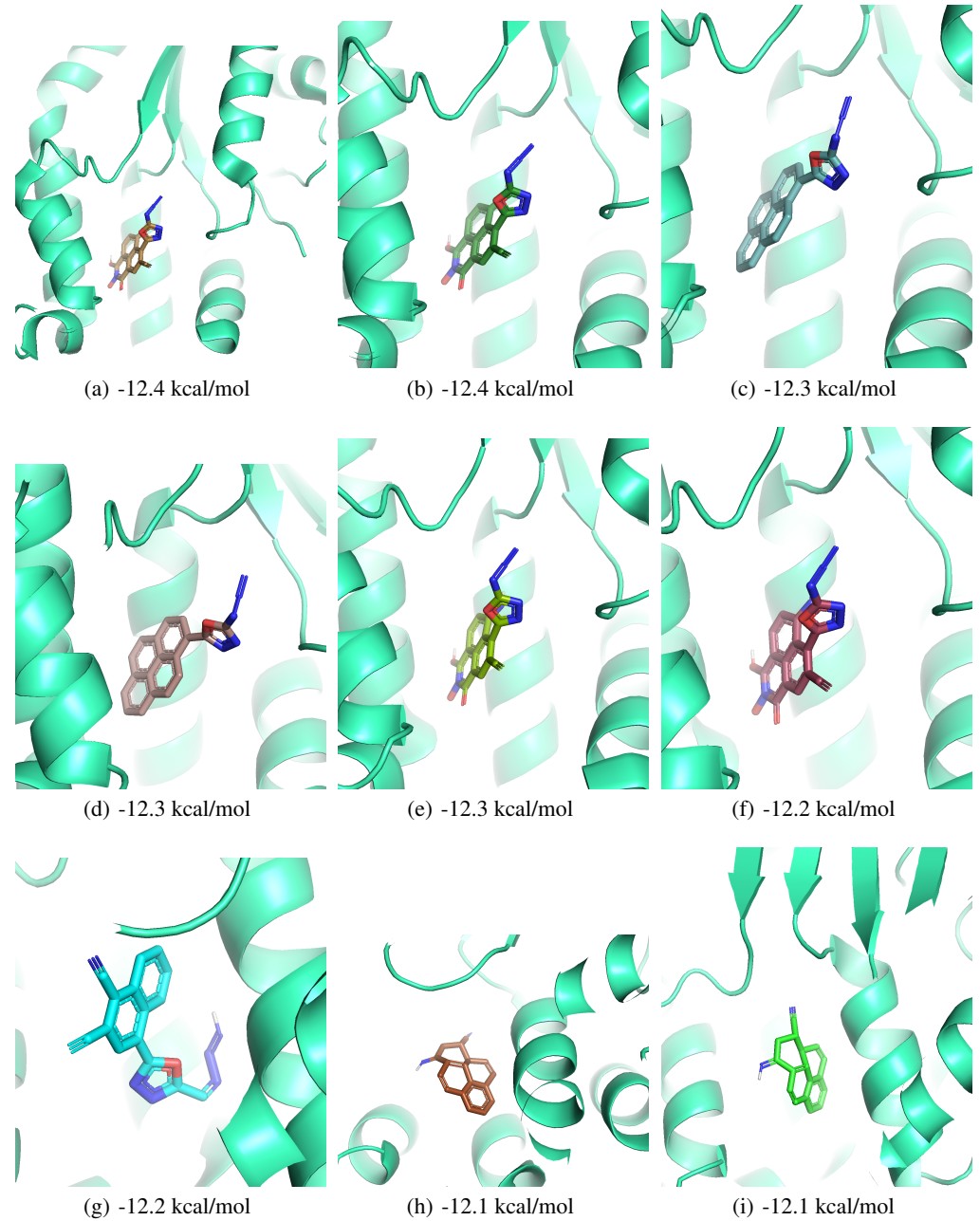

(a) -12.4 kcal/mol  (b) -12.4 kcal/mol  (c) -12.3 kcal/mol

(d) -12.3 kcal/mol  (e) -12.3 kcal/mol  (f) -12.2 kcal/mol

(g) -12.2 kcal/mol  (h) -12.1 kcal/mol  (i) -12.1 kcal/mol

Figure 9: Example of ligand poses (generated by RGA) and binding sites of target structures "4unn".