# OpenReview forum: "Reinforced Genetic Algorithm for Structure-based Drug Design"
_NeurIPS.cc/2022/Conference — NeurIPS 2022 Accept_

### Official Review · Reviewer_S7k1 · 2022-07-10

**Rating:** 4
**Confidence:** 5
**Soundness:** 3 good
**Presentation:** 2 fair
**Contribution:** 2 fair

**Summary:**

This authors of  this submission propose to apply structure based drug design information to the selection, crossover, and mutation operations of evolutionary computation for drug discovery. An equivariance neural network is used to represent target-ligand binding complex and used as a base network for crossover and mutation action selections. In these evolutionary operations, substructure similarity was considered in crossover; and mutation was inspired by chemical reactions. This proposed method is a hybrid model combining deep reinforcement learning and evolutionary optimization for drug design. Experimental results show that the proposed method outperform existing benchmarks. Ablation studies show that the incorporation of structural information in crossover and mutation can improve performance in terms of several metrics.

**Questions:**

1. I would like to see the performance of binding affinity regression in ENNs.
2. I would like to see more discussions of hybrid models, such as RL+EC, DL+EC, etc.
3. Significance studies, in terms of p-values, should be reported to compare the proposed work with existing (naive) GA algorithms for drug design.
4. Scalability of this proposed model should be explicitly discussed as well.

**Ethics Review Area:**

["I don’t know"]

**Limitations:**

Limitations and potential improvements of the work were slightly discussed in the conclusion section.

**Strengths And Weaknesses:**

Strengths:
1. The consideration of structural information through policy networks for selection, crossover, and mutation are the major contribution of this work from my viewpoint. The combination of reinforcement learning (RL) and evolutionary computation (EC) may be novel for this application. However, the authors should discuss more about existing work that combines RL and EC in general.
2. Experimental benchmarkings are thorough. Various well-known existing models (modified to integrate docking) were compared.

Weaknesses:
1. The regression performance of the ENNs was not discussed. The performance of all other operations depends on the reliability of such ENNs networks. Thus, the authors must show their regression results in pretraining ENNs.
2. Evolutionary operations were largely influenced by Autogrow 4.0. The major difference is that policy networks were employed. The base neural networks used in this work are existing architectures, which weakens the novelty of this work. The signifiance of the result obtained by the proposed work (which is a complicated process) in comparison with other simpler GAs should be further studies, in terms of p-value.
3. The presentation of this paper could be improved. Line 230: is X a mutation child? a MLP -> an MLP. structured based -> structure based. References should be reformatted. Please pay attention of capitalizations.

---

> ### Author Response · Authors · 2022-08-02
> **Our method significantly outperforms other GA methods;  more discussions on related works are added.**
>
> Thanks for your comments. We also highlight the changes in the draft in red.
>
> **Q1**: Performance of binding affinity regression in ENNs.
>
> **A1**: We add validation loss of pre-training ENN on target-ligand binding affinity prediction task and other setups in Fig. 5 and Section F.2 in Appendix. We find the learning process (validation loss) converges within an epoch. Validation loss function is root mean square error (RMSE).
>
> **Q2**: More discussions on RL+EC, DL+EC.
>
> **A2**: We add the following discussion of related works on combinatorial optimization (CO), evolutionary computation (EC), reinforcement learning (RL) and deep learning (DL) in Section G of Appendix.
>
> - (RL+CO:) Traditional CO is essentially a searching problem in discrete space, and RL has been widely used to enhance CO. For example, AlphaGo [1] trains neural networks to estimate state values and uses it to prioritize the searching branches within the framework of traditional Monte Carlo tree search (MCTS). In the field of molecular design, similar approaches have been developed to guide tree search in growing molecular branches, e.g., MolDQN [2] & GCPN [3]. But all of the above examples focus on tree search formulation of the problem, which has been demonstrated to be less efficient in optimization [4, 5].
>
> - (DL/RL+EC:) EC is a special kind of CO that maintains a population of solutions each step and recombines the information between the solutions in population, which is the key to EC’s superior performance, as validated in several molecule optimization benchmarks [4, 5]. However, training neural network to enhance the performance is not a trivial extension due to the lack of a suitable problem formulation. GEGL [6] adds LSTM generation as one genetic process to enhance the performance of GA. A straightforward approach is to train a network and combine it with fitness evaluation. For example, [7] trains a neural network-based property predictor and incorporates it to make a model-based evolutionary algorithm. GA+D[8] leverages neural network as a discriminator to measure the drug’s proximity to the training data, which is incorporated as a scorer in fitness evaluation.
>
> **Q3**: Significance studies (p-value).
> **A3**: We conduct hypothesis testing to show the statistical significance of our method over other GA-based methods. Specifically, we compare the significance of the improvement over GA methods (AutoGrow4, Graph-GA, GA+D, GEGL) via running 5 independent trials with different random seeds and then evaluating p-value. We consider top-100/10/1 scores, the most important metrics for optimization performance.
>
> |  | p-value on TOP-100 | p-value on TOP-10 | p-value on TOP-1 |
> | ------------------ | ----- | ----- | ------ |
> | RGA v.s. AutoGrow 4 | 0.002 | 0.005 | 0.07  |
> | RGA v.s. Graph-GA |  0.003 | 0.010 | 0.046  |
> | RGA v.s. GA+D |  1.0e-7 | 5.0e-5 | 3e-4  |
> | RGA v.s. GEGL | 2.5e-4 | 3.7e-3 | 5.0e-3 |
>
> We find almost all the p-values are less than 0.05 (except one value), indicating the improvements of RGA over other GA methods are statistically significant. Details are in Section F.5 in Appendix.
>
>
> **Q4**: Scalability.
>
> **A4**: We report computing time of all the methods in Fig.4 and add scalability analysis as followed in Section F.4 in Appendix:
>
> “Our method has 2 operations: crossover & mutation. Section E.2 shows the population size is K=50, the chemical reaction set is R, |R|=94.
>
> - crossover
>   - (1) select the first parent molecule and evaluate the probability over all the molecules in the current population, complexity is O(K).
>   - (2) select the second parent molecule based on the first parent molecule and evaluate the probability over all the remaining molecules in the current population (Eq. 2), complexity is O(K-1). The whole complexity is O(K).
>
> - Mutation
>   - (1) select the parent molecule (only 1 parent) via evaluating the probability over all the molecules in the current population, complexity is O(K).
>   - (2) select a mutated molecule (generated by chemical reaction) (Eq. 3). The complexity is O(|R|), where R is the reaction set. Complexity is O(|R|+K).
>
> Overall, our method scales linearly with population size and size of the chemical reaction set.”
>
> **Q5**: Presentation.
>
> **A5**: We improve it following your suggestions.
>
> Ref
> ===
>
> 1 Mastering the game of Go with deep neural networks and tree search. Nature16.
>
> 2 Optimization of molecules via deep reinforcement learning. Sci. Rep. 19.
>
> 3 Graph convolutional policy network for goal-directed molecular graph generation. NeurIPS18.
>
> 4 GuacaMol: benchmarking models for de novo molecular design. JCIM19.
>
> 5 Sample Efficiency Matters: A Benchmark for Practical Molecular Optimization. arXiv22.
>
> 6 Guiding deep molecular optimization with genetic exploration. NeurIPS20.
>
> 7 Evolutionary design of molecules based on deep learning and genetic algorithm. Sci. Rep. 21.
>
> 8 Augmenting genetic algorithms with deep neural networks for exploring the chemical space. ICLR20.

---

### Official Review · Reviewer_KpKV · 2022-07-11

**Rating:** 3
**Confidence:** 4
**Soundness:** 2 fair
**Presentation:** 2 fair
**Contribution:** 2 fair

**Summary:**

The paper proposes a Reinforced Genetic Algorithm that combines different models for structure-based drug design. The algorithm takes the 3D structure of the targets and ligands as inputs and is pre-trained using native complex structures to utilize the knowledge of the shared binding physics from different targets and then fine-tuned during optimization. Empirical studies show that the proposed method outperforms the baselines and the ablation study suggests the contribution of two training strategies.

**Questions:**

I would suggest the author focus on one aspect of the method instead of considering multiple model combinations. This creates confusion on which part is more important than others.

**Limitations:**

See above

**Strengths And Weaknesses:**

Strength: The paper proposes a new method pipeline to solve an existing drug design problem.

Weakness: There is very little technical innovation, more like a direct application of existing methods to a specific problem. Regarding the model design, there is no explanation of the rationalities. For example, why did the authors combine the MDP with a EE neural network? There is also no explanation of the theoretical support for convergence or a better performance guarantee. In addition, the method is heavily adapted to one specific problem and can’t be generalized to a wide range of problems. The experiments are not representative either.

---

> ### Author Response · Authors · 2022-08-02
> **Our method is the first attempt to use reinforcement learning to suppress random behavior in genetic algorithm**
>
> Thank you for your comments. Please find our response below. We highlighted the changes in the draft in red.
>
> **Q1**: There is very little technical innovation, more like a direct application of existing methods to a specific problem.
>
> **A1**: As far as we know, our paper is the first to propose a general framework that reinforcement learning (RL) to prioritize the promising actions to suppress the random behavior in traditional genetic algorithms (GA). Our method is significantly different from existing DL(deep learning)/RL+GA methods: GA+D [1] trains a discriminator to enhance genetic algorithms, and GEGL [2] learns LSTM-based hill climbers to imitate the behavior of genetic algorithms. If you find an existing method closer to what we have done, we would like to acknowledge it in our paper.
>
> **Q2**: Regarding the model design, there is no explanation of the rationalities. For example, why did the authors combine the MDP with a EE neural network?
>
> **A2**: We have explained our design rationale for the algorithm in the Introduction and the Method sections. We emphasize here that our primary rationale is to train a neural network for exploration guidance to suppress the random walk behavior of genetic algorithms and utilize the proteins' geometric information (line 56-72); Equivariant neural network is the state-of-the-art neural architecture for geometric modeling. It is a natural choice to utilize equivariant neural networks to represent the geometric structure of the target-ligand complex.
>
> **Q3**: In addition, the method is heavily adapted to one specific problem and can’t be generalized to a wide range of problems. The experiments are not representative either.
>
> **A3**: The proposed reinforced genetic algorithm is a general framework that can be used in other combinatorial optimization problems, e.g., symbolic regression [3] and ​​quantum circuit design [4], as discussed in the Conclusion section.
>
> **Q4**: I would suggest the author focus on one aspect of the method instead of considering multiple model combinations. This creates confusion on which part is more important than others.
>
> **A4**: We believe algorithms are proposed to solve problems. For example, Rainbow [5] is a combination of multiple improvements in value learning reinforcement learning, but it demonstrates the combination is helpful but not harmful, which is not trivial and impactful. In this paper, we focused on proposing a new algorithm that achieved superior performance but not simply comparing RLs and GAs to show which part is more important. We also conducted ablation studies to explore the effect of these two policy networks and reported the results in Section F.3 and Table 3. The results reveal that the crossover policy network is more critical than the mutation policy network in optimization performance.
>
>
> Reference
> ===
>
> [1] Nigam et al. Augmenting genetic algorithms with deep neural networks for exploring the chemical space. ICLR 2020.
>
> [2] Ahn et al. Guiding deep molecular optimization with genetic exploration. NeurIPS 2020.
>
> [3] Michael Schmidt and Hod Lipson. Distilling free-form natural laws from experimental data. science, 2009.
>
> [4] Yuxuan Du, Tao Huang, Shan You, Min-Hsiu Hsieh, and Dacheng Tao. Quantum circuit architecture search: error mitigation and trainability enhancement for variational quantum solvers. arXiv 2020.
>
> [5] Hessel, Matteo, et al. "Rainbow: Combining improvements in deep reinforcement learning." AAAI. 2018.

---

### Official Review · Reviewer_QPzy · 2022-07-11

**Rating:** 6
**Confidence:** 4
**Soundness:** 3 good
**Presentation:** 3 good
**Contribution:** 2 fair

**Summary:**

This paper proposes a genetic algorithm (mostly based on Autogrow) guided by reinforcement learning. The RL agent in this method can directly perceive the 3D structure of protein targets. In contrast, Autogrow searches molecules in a ``blind’’ way, unable to directly see the 3D structure during sampling.

In this work, RL is used to guide two processes in the genetic algorithm: crossover and mutation. The crossover agent selects parents and mutation for producing ``children’’ molecules. The mutation agent selects chemical reaction to modify the molecules.

This paper presented a very comprehensive evaluation of the proposed method and existing SBDD methods. The improvement over of the method over previous ones is evidenced by the comprehensive experiment, though it is not very significant compared to traditional genetic algorithm.

**Questions:**

See strengths and weakness.

**Limitations:**

See strengths and weakness.

**Strengths And Weaknesses:**

Strengths

(1) Using RL to empower GA-based SBDD is a good idea. It can make the GA process more directed and uses more information such as protein 3D structures.

(2) Extremely comprehensive survey and evaluation of previous SBDD methods. The survey and evaluation cover a wide range of methods including graph-based, 3D-based, and GA-based.

(3) Acceptable performance improvement. Though the improvement in performance is over GA-based algorithms is not that significant, it surpasses GNN-based and 3D-based methods by a very significant margin.

Weakness

(1) Bias in evaluation metric. The algorithm optimizes Vina score, it is not surprising that the Vina score is good in evaluation. Have the author considered using different scoring functions for GA and testing?

(2) Few examples of generation. I did not find any presentation of generated molecules in the main text or in the supplementary material. Examples are important and their comparison to a ``ground truth’’ molecules are important to understand and evaluate what the model generates.

---

> ### Author Response · Authors · 2022-08-02
> **We clarify the usage of Autodock Vina, and add more examples.**
>
> Thank you for your comments. Please find our response below. We highlighted the changes in the draft in red.
>
> **Q1**: Bias in evaluation metric. The algorithm optimizes Vina score, it is not surprising that the Vina score is good in evaluation. Have the authors considered using different scoring functions for GA and testing?
>
> **A1**: We use Autodock Vina score as the main objective because it is a mainstream optimization metric in structure-based drug design [1, 2] and open-sourced [3]. We acknowledge the defect of Vina scores, but the focus of our paper is proposing a new structure-based drug design method. Users can adopt our method to other scores they want without changing the algorithm as long as they have access to, such as Glide, a commercialized docking score [4, 5].
>
> **Q2**: Few examples of generation. I did not find any presentation of generated molecules in the main text or in the supplementary material. Examples are important and their comparison to a “ground truth” molecules are important to understand and evaluate what the model generates.
>
> **A2**: In the submission, we had two examples in Figure 2a and 2b. In the revision, we add more examples and descriptions in Section F.6 in Appendix. We find the generated ligands are binding tightly with the target proteins.
>
>
> Reference
> ===
>
> [1] Jacob O Spiegel and Jacob D Durrant. Autogrow4: an open-source genetic algorithm for de
> novo drug design and lead optimization. Journal of cheminformatics, 12(1):1–16, 2020.
>
> [2] Shitong Luo, Jiaqi Guan, Jianzhu Ma, and Jian Peng. A 3D generative model for structure-based drug design. NeurIPS, 2021.
>
> [3] https://vina.scripps.edu
>
> [4] Lyu, Jiankun, et al. "Ultra-large library docking for discovering new chemotypes." Nature 566.7743 (2019): 224-229.
>
> [5] Halgren, Thomas A., et al. "Glide: a new approach for rapid, accurate docking and scoring. 2. Enrichment factors in database screening." Journal of medicinal chemistry 47.7 (2004): 1750-1759.

---

> > ### Comment · Reviewer_QPzy · 2022-08-09
> > **Re: We clarify the usage of Autodock Vina, and add more examples.**
> >
> > Thank you for your response!
> >
> > Agree that the paper focuses on developing a new SBDD algorithm.
> > What I mean is that, it would make the paper stronger if the author optimizes the molecule with Vina, and evaluates them using another docking software such as Glide (or others).
> > Using another independent evaluation docking software would be more convicing, but I think the current setting is also reasonable.
> >
> > I appreciate the additional examples. Thanks.

---

### Official Review · Reviewer_yDTi · 2022-07-17

**Rating:** 7
**Confidence:** 3
**Soundness:** 3 good
**Presentation:** 3 good
**Contribution:** 3 good

**Summary:**

This article introduces a reinforced genetic algorithm to tackle the problem of structure-based drug design, one of the key approaches to achieve fast and efficient drug discovery. The authors overcome limitations of current genetic algorithms, which is one of the most efficient screening procedures for binding affinity, by reformulating it as a Markov decision process, using neural networks to make informed decisions and suppress the random-walk behavior.

**Questions:**

In the diversity metric, the RGA algorithm introduced in this paper provides somewhat worse results than competing methods and even brute force screening. Could the authors comment on the importance of this metric and whether some improvements could be made in this respect?


**Ethics Review Area:**

["Responsible Research Practice (e.g., IRB, documentation, research ethics)"]

**Limitations:**

While this work provides a useful approach to prioritizing promising drug design steps, the pathway to an implementation in real life drug design cycle remains unclear.

**Strengths And Weaknesses:**

This article is well written and provides meaningful theoretical advances to a key pharmaceutical science problem. It provides thorough comparisons with current existing methods.

---

> ### Author Response · Authors · 2022-08-02
> **RGA can be seamlessly used in drug design, diversity is an auxiliary metric.**
>
> Thank you for your comments. Please find our response below. We highlighted the changes in the draft in red.
>
>
> **Q1**: Could the authors comment on the importance of diversity and whether some improvements could be made in this respect?
>
> **A1**: Diversity is an auxiliary metric of molecule optimization compared with Vina score (measuring binding affinity between target and ligand). The goal of using diversity is to ensure the method is not heavily exploiting a small portion of the chemical space. As we observe from Table 1, all the methods except rationaleRL achieve over 0.8 diversity score, which indicates diversity is not a big problem right now. We acknowledge the limitation of Tanimoto similarity, but proposing another diversity measurement is not the focus of this paper.
>
>
> **Q2**: While this work provides a useful approach to prioritizing promising drug design steps, the pathway to an implementation in real life drug design cycle remains unclear.
>
> **A2**: As our method is a direct improvement from a traditional structure-based drug design method, it could be seamlessly adopted for hit discovery at the beginning of a real-life drug design or lead optimization once we have a suboptimal hit. In both cases, our method could propose promising candidate chemicals ready for experimental evaluation as a next step.

---

### Meta-Review · Area_Chair_72qo · 2022-08-26

**Recommendation:** Accept
**Confidence:** Certain

**Metareview:**

This paper proposes a genetic algorithm guided by reinforcement learning to design molecules. The RL agent in this method can directly perceive the 3D structure of protein targets. This paper presented a very comprehensive evaluation of the proposed method and existing SBDD methods. The improvement over of the method over previous ones is evidenced by the comprehensive experiment, though it is not very significant compared to traditional genetic algorithm.
Authors may want analyze the biases and show a few examples of generated molecules to better illustrate the effectiveness.

**Award:**

No

---

### Decision · Program_Chairs · 2022-09-14

Accept